# Beurling-Selberg extremization and modular bootstrap at high energies

**Baur Mukhametzhanov and Sridip Pal**

Institute for Advanced Study, Princeton, NJ 08540, U.S.A.

## Abstract

We consider previously derived upper and lower bounds on the number of operators in a window of scaling dimensions $[\Delta - \delta, \Delta + \delta]$ at asymptotically large $\Delta$ in 2d unitary modular invariant CFTs. These bounds depend on a choice of functions that majorize and minorize the characteristic function of the interval $[\Delta - \delta, \Delta + \delta]$ and have Fourier transforms of finite support. The optimization of the bounds over this choice turns out to be exactly the Beurling-Selberg extremization problem, widely known in analytic number theory. We review solutions of this problem and present the corresponding bounds on the number of operators for any $\delta \geq 0$. When $2\delta \in \mathbb{Z}_{\geq 0}$ the bounds are saturated by known partition functions with integer-spaced spectra. Similar results apply to operators of fixed spin and Virasoro primaries in $c > 1$ theories.

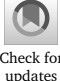

# 1   Introduction

Crossing equations in Conformal Field Theories (CFT) form a system of infinite number of equations on an infinite number of unknowns: the spectrum and OPE coefficients. Arguably, the simplest way of extracting constraints on conformal data from crossing equations, is to take an (euclidean) OPE limit in one of the channels, call it t-channel. In this limit t-channel is typically dominated by the vacuum or another light state. Next, one finds a density of high-energy states or OPE coefficients in another channel, call it s-channel, that reproduces the vacuum contribution in the t-channel in the OPE limit. This density is sometimes called a "crossing kernel". Many examples of this type for various crossing equations have been discussed in the literature [1–13].

This approximation, when we keep only the lightest state in the t-channel, is extremely useful for obtaining certain coarse-grained features of the high-energy spectrum. Prominently, thermodynamic behavior of high-energy states and aspects of classical geometry of a gravity dual [14] can be understood in this way.

On the other hand, many interesting fine-grained features of the high-energy spectrum are not captured in this approximation. One very basic and important example is discreteness of energy eigenstates. The coarse-grained approximation discussed above typically predicts a continuous spectrum. This is akin to a version of the information paradox in AdS [15] and related to the late-time behavior of correlators [16–19] and the spectral form-factor [20, 21]. Another important example is the Random Matrix Theory (RMT) behavior of energy eigenstates. Just like coarse-grained thermality, RMT is expected to possess a certain degree of universality. One expects that it holds in a broad class of "chaotic" theories. A natural conjecture is that in any CFT without conserved currents except for the stress tensor, the energy eigenstates at asymptotically high energies obey RMT statistics.

In an attempt to derive universal fine-grained features of CFT spectra, such as RMT, one might imagine a two-step strategy. First, we need to understand why the coarse-grained approximation is not enough and quantify its limitations. Second, we would like to make extra assumptions (e.g. absence of conserved currents) that restrict us to chaotic theories. One might expect that these assumptions lift the limitations of the coarse-grained approximation and allow us to probe fine-grained features of the spectrum.

Partial progress on the step one has been made in [22–28]. The goal of the present work is to report on further progress in this direction. We do not have anything to say about the step two at this time.

In this paper we consider unitary 2d CFTs with a modular invariant partition function

$$Z(\beta) = \sum_\Delta e^{-\beta\left(\Delta - \frac{c}{12}\right)} = \sum_\Delta e^{-\frac{4\pi^2}{\beta}\left(\Delta - \frac{c}{12}\right)} \, . \tag{1}$$

At high temperatures it has an asymptotic behavior

$$\int_0^\infty d\Delta\, \rho(\Delta) e^{-\beta\Delta} \approx e^{\frac{4\pi^2}{\beta}\frac{c}{12}} + e^{\frac{4\pi^2}{\beta}\left(\frac{c}{12}-\Delta_1\right)} + \dots\,, \quad \beta \to 0\,, \tag{2}$$

$$\rho(\Delta) = \sum_{\Delta_n} \delta(\Delta - \Delta_n)\,. \tag{3}$$

From this one would like to derive the Cardy formula

$$\rho(\Delta) \stackrel{?}{\sim} \exp\left(2\pi\sqrt{\frac{c\Delta}{3}}\right), \qquad \Delta \to \infty\,. \tag{4}$$

To do that one is tempted to take the inverse Laplace transform of the vacuum contribution (2)

$$\rho(\Delta) = \int_{-\infty}^{\infty} \frac{dt}{2\pi} e^{it\left(\Delta - \frac{c}{12}\right)} Z(\epsilon + it) \tag{5}$$

$$= \int_{\epsilon-i\infty}^{\epsilon+i\infty} \frac{d\beta}{2\pi i}\, e^{\beta\left(\Delta - \frac{c}{12}\right)} \left( e^{\frac{4\pi^2}{\beta}\frac{c}{12}} + e^{\frac{4\pi^2}{\beta}\left(\frac{c}{12}-\Delta_1\right)} + \dots \right). \tag{6}$$

If we kept only the first term in parentheses, i.e. the vacuum contribution, we would indeed get Cardy growth (4) [1]. However, the vacuum contribution dominates only for $|\mathrm{Im}(\beta)| \ll 1$, while for $|\mathrm{Im}(\beta)| \gtrsim 1$ we lose control over the integrand. In the latter region contributions of all operators in (6) become of $O(1)$. Naively, one would like to argue that in the limit $\Delta \to \infty$ only the region $|\mathrm{Im}(\beta)| \lesssim 1/\Delta$ gives a significant contribution. Such intuition is usually based on Riemann-Lebesgue lemma. In this case it is not applicable because the function $Z(\epsilon + it)$ is not integrable.[2]

In a theory with discrete spectrum the density of states $\rho(\Delta)$ is a sum of delta-functions and, therefore, we expect the integral (5) to be very sensitive to the precise value of $\Delta$ even at large $\Delta$: it's either zero or infinite. To compute this fine-grained density one would need to know much more than just the vacuum operator.

Of course, what is really meant by (4) is the coarse-grained density, i.e. the density of states averaged over a small window around $\Delta$. One may consider different ways to average, for example

$$\int_{\Delta-\delta}^{\Delta+\delta} d\Delta'\rho(\Delta') \sim \#\exp\left(2\pi\sqrt{\frac{c\Delta}{3}}\right), \quad \Delta \to \infty\,. \tag{7}$$

The question then remains: how does one derive (7)? This was explained in [25], where a formula analogous to (5) was derived for the averaged density of states, but with an insertion of certain kernels $\widehat{\phi}_\pm(t)$. The role of $\widehat{\phi}_\pm(t)$ is to cutoff the troubling region $t \to \infty$, where one loses control over the integrand. This effectively localizes the integral to the region of sufficiently small $t$, where the integral is controlled by the vacuum contribution. The price to pay for this modification is that instead of equality we get upper/lower bounds (hence "$\pm$" in $\widehat{\phi}_\pm(t)$) on the averaged density of states. This will be reviewed in section 2.

It was shown in [25] that a simple sufficient choice of $\widehat{\phi}_\pm(t)$ is to require that they have finite support $|t| < 2\pi$. In this paper we find the optimal functions $\widehat{\phi}_\pm(t)$ of this type. The

---

[1] Up to a controllable divergence that gives a delta-function.

[2] This is clear, for example, from the fact that there are recurrences. In particular, in theories with integer-spaced spectra, that we will discuss, the recurrences are perfect and $Z(\epsilon + it)$ has an infinite number of peaks, where it takes the same value as at $t = 0$.

problem boils down to finding functions $\phi_\pm(x)$ that majorize/minorize a characteristic function of an interval $\theta_{[-\delta,\delta]}(x)$ and minimize $L^1$ norm $||\phi_\pm - \theta_{[-\delta,\delta]}||$ with a constraint that Fourier transformations $\widehat{\phi}_\pm(t)$ have finite support $|t| < 2\pi$. This turns out to be a classic Beurling-Selberg problem [29, 30], widely known in analytic number theory.

In particular, this allows us to derive a simple bound

$$(2\delta - 1)\rho_0(\Delta) \leq \int_{\Delta-\delta}^{\Delta+\delta} d\Delta' \rho(\Delta') \leq (2\delta + 1)\rho_0(\Delta), \qquad \delta \geq 0 \, , \Delta \to \infty \, , \tag{8}$$

where $\rho_0$ is defined in (22). The bounds (8) need some clarification. By the limit $\Delta \to \infty$ we mean that both upper and lower bounds have corrections that can be either positive or negative, but suppressed in $\Delta$. More rigorously, the bounds take the form[3]

$$2\delta - 1 \leq \liminf_{\Delta\to\infty} \frac{N_{\delta-0}(\Delta)}{\rho_0(\Delta)} \leq \limsup_{\Delta\to\infty} \frac{N_{\delta+0}(\Delta)}{\rho_0(\Delta)} \leq 2\delta + 1 \, , \tag{9}$$

where we defined the number of states in the interval

$$N_\delta(\Delta) = \int_{\Delta-\delta}^{\Delta+\delta} d\Delta' \rho(\Delta') \, . \tag{10}$$

In (9) we were also careful to write $\delta \pm 0$, indicating whether we include on exclude the states at the edges $\Delta \pm \delta$. The origin of this will be discussed in section 2. One can think of $N_\delta(\Delta)$ as a "staircase" function oscillating around its average value. Therefore, the appearance of $\limsup$ and $\liminf$ instead of $\lim$ is natural in (9).

Throughout the paper we will mostly write the bounds in the form (8) for brevity, but one should keep in mind that the rigorous form that is implied is given by (9).

It turns out that the bounds (8) are optimal, among those obtained from bandlimited functions $\phi_\pm(x)$, only when $2\delta \in \mathbb{Z}$. In this case they are also saturated by $c = 4k, k \in \mathbb{Z}_{>0}$ partition functions with integer spaced spectra, for example, Klein's $j$-invariant. We derive these results for $2\delta \in \mathbb{Z}$ in section (3). In section 4 we derive (8) for any $\delta \geq 0$. The optimal bounds for $2\delta \notin \mathbb{Z}$ are derived in section 5.

Note the following two simple consequences of (8). The lower bound implies that:

1)*In any window of size $2\delta > 1$ at asymptotically high energies there is non-zero number of operators.*

This result was previously established in [25, 26]. If we have Virasoro symmetry this is trivial due to descendants.[4] But the whole analysis can be repeated almost verbatim for Virasoro primaries (when $c > 1$) and the result remains essentailly the same, as we show in section 7.

The upper bound in the limit $\delta \to 0$ becomes $\rho_0(\Delta)$. Therefore:

2)*The maximum degeneracy of an individual operator with dimension $\Delta$ is $\rho_0(\Delta)$ up to additive corrections suppressed at asymptotically high energies.*

This is again saturated by partition functions with integer-spaced spectra for $c = 4k, k \in \mathbb{Z}_{>0}$, as we will discuss in section 3.3.

We generalize our results to operators of fixed spin in section 6 and Virasoro primaries with arbitrary or fixed spin in $c > 1$ theories in section 7.

---

[3]Recall the definition $\limsup_{x\to\infty} f(x) = \lim_{y\to\infty} \sup_{x>y} f(x)$ and similarly for $\liminf$.

[4]Note, however, that all we require for (8) to be true is (1). Therefore, we need only scaling symmetry and not full conformal symmetry.

The main results of this paper are the upper and lower bounds (8), (21), (83), (84), and similar formulas for fixed spin operators (105), (110), (111), (112). These bounds are optimal among those that can be obtained using bandlimited functions $\phi_\pm(x)$. We do not know whether the bounds can be saturated for central charges $c \neq 4k, k \in \mathbb{Z}_{>0}$ or for $2\delta \notin \mathbb{Z}$. Similar results apply to Virasoro primaries (127), (129).

## 2   Review

In this section we review the setup of [25] and make a few additional comments. We start with defining two continuous integrable functions $\phi_\pm(\Delta')$ that bound the indicator function of an interval $\Delta - \delta < \Delta' < \Delta + \delta$

$$\phi_-(\Delta') \le \theta_{(\Delta-\delta,\Delta+\delta)}(\Delta') \le \theta_{[\Delta-\delta,\Delta+\delta]}(\Delta') \le \phi_+(\Delta') \,. \tag{11}$$

Here we have been careful with the ends of the interval. The function $\theta_{(\Delta-\delta,\Delta+\delta)}$ vanishes at the ends, while $\theta_{[\Delta-\delta,\Delta+\delta]}$ at the ends is 1. Since the functions $\phi_\pm$ are continuous, we can think of $\phi_+$ as a bound from above where we include the edges, while $\phi_-$ is a bound from below where we do not include the edges, see the figure[5] 1. This will be important later, when we check optimality of our bounds. It will correspond to whether we include or not the states at the edges. Until then we will not distinguish the two $\theta$'s for the sake of brevity.

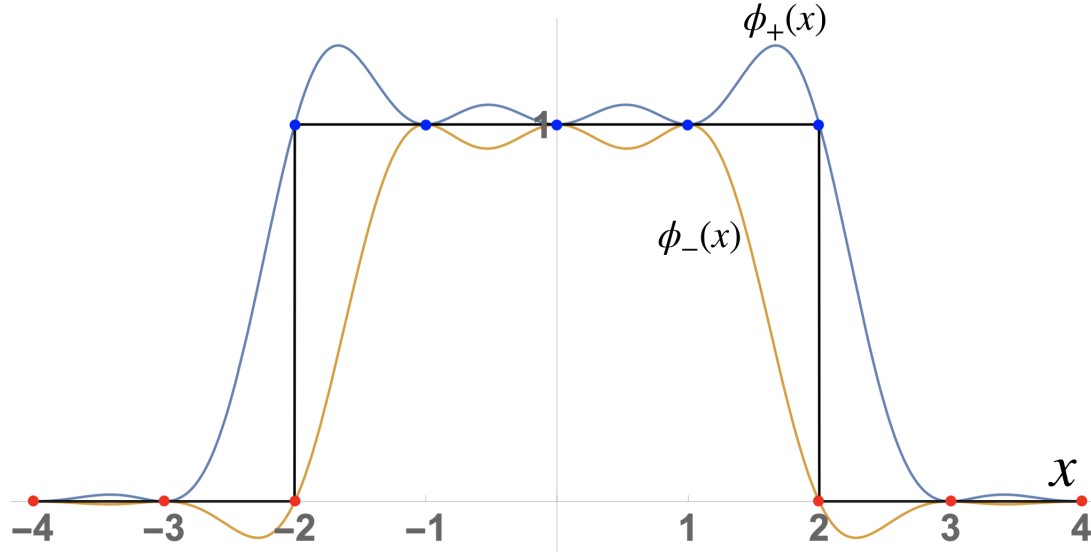

Figure 1: Functions $\phi_\pm(x)$ majorising and minorising the indicator function of the interval $[-2, 2]$.

Multiplying (11) by exponential factors[6] $e^{\Delta\pm\delta-\Delta'}$ and integrating with the density of states

---

[5]We are being somewhat cavalier about the argument of the functions $\phi_\pm$. We use both $\phi_\pm(\Delta')$ and $\phi_\pm(x)$ with $x = \Delta' - \Delta$ interchangeably.

[6]We can do this because for $|\Delta' - \Delta| \le \delta$ we have $e^{\Delta-\delta-\Delta'} \le 1 \le e^{\Delta+\delta-\Delta'}$ and for $|\Delta' - \Delta| > \delta$ the inequality (11) takes the form $\phi_-(\Delta') \le 0 \le \phi_+(\Delta')$, so we can multiply by positive factors $e^{\Delta\pm\delta-\Delta'}$.

$\rho(\Delta')$ we find

$$e^{\beta(\Delta-\delta)} \int_{-\infty}^{\infty} d\Delta' \, \rho(\Delta') e^{-\beta\Delta'} \phi_-(\Delta')$$

$$\leq \int_{\Delta-\delta}^{\Delta+\delta} d\Delta' \rho(\Delta') \leq \tag{12}$$

$$e^{\beta(\Delta+\delta)} \int_{-\infty}^{\infty} d\Delta' \, \rho(\Delta') e^{-\beta\Delta'} \phi_+(\Delta') \, .$$

Taking the Fourier transform of $\phi_\pm$, one finds the bounds on the number of states [25]

$$e^{\beta(\Delta-\delta-c/12)} \int_{-\infty}^{\infty} dt \, Z(\beta+it)\widehat{\phi}_-(t) e^{-itc/12}$$

$$\leq \int_{\Delta-\delta}^{\Delta+\delta} d\Delta' \rho(\Delta') \leq \tag{13}$$

$$e^{\beta(\Delta+\delta-c/12)} \int_{-\infty}^{\infty} dt \, Z(\beta+it)\widehat{\phi}_+(t) e^{-itc/12} \, ,$$

where $\widehat{\phi}_\pm(t)$ is the Fourier transform of $\phi_\pm(\Delta')$

$$\phi_\pm(\Delta') = \int dt \, e^{-it\Delta'} \widehat{\phi}(t) \, . \tag{14}$$

The integrals in (13) are reminiscent of the inverse Laplace transform (5) in the sense that we integrate the partition function over imaginary inverse temperatures. However, in comparison wtih (5), here we have much more control over the integrand. In particular, we can choose the functions $\phi_\pm(\Delta')$ in such a way that the region $t \gtrsim 1$, where we lose control over the integrand, does not contribute. This leads us to consider $\phi_\pm(\Delta')$ such that their Fourier $\widehat{\phi}_\pm(t)$ have finite support $|t| < \Lambda$. Functions of this type are called *bandlimited functions*.

We would like to bound the number of states in (13) at large $\Delta$. In this case we imagine $\beta$ to be small. Later we will see that to optimize the bounds $\beta$ should be related to $\Delta$ by the standard thermodynamic relation.

The quantity $|Z(\beta+it)|^2$ is called the spectral form-factor [20, 21]. Its typical behavior is that it is large at $t = 0$ and decays exponentially at small times $t$ due to oscillating phases. Early times are controlled by the vacuum in the dual channel [21]. For a large enough $t = t_{rec}$ phases can come into sync again and a recurrence happens, when the form-factor is again large. See figure 2. The recurrence time and the value of the form-factor at the recurrence time depend on the particular theory we study. Therefore, we expect that the integrand in (13) is controlled by the vacuum in the dual channel only for $t \lesssim t_{rec}$. This suggests that we should take $\Lambda \lesssim t_{rec}$.

For partition functions with integer-spaced spectra, such as Klein's $j$-invariant, the recurrence time is $t_{rec} = 2\pi$. In fact, as was shown in [25] and will be reviewed below, in any 2d CFT for $\Lambda \leq 2\pi$ the integrals in (13) are dominated by the vacuum in the S-dual channel. Therefore, $t_{rec} = 2\pi$ is the shortest recurrence time among modular invariant partition functions.

Using modular invariance and splitting the partition function into light (below $c/12$) and heavy (above $c/12$) operators, we find the upper bound in the limit $\Delta \to \infty, \beta \to 0$ (lower

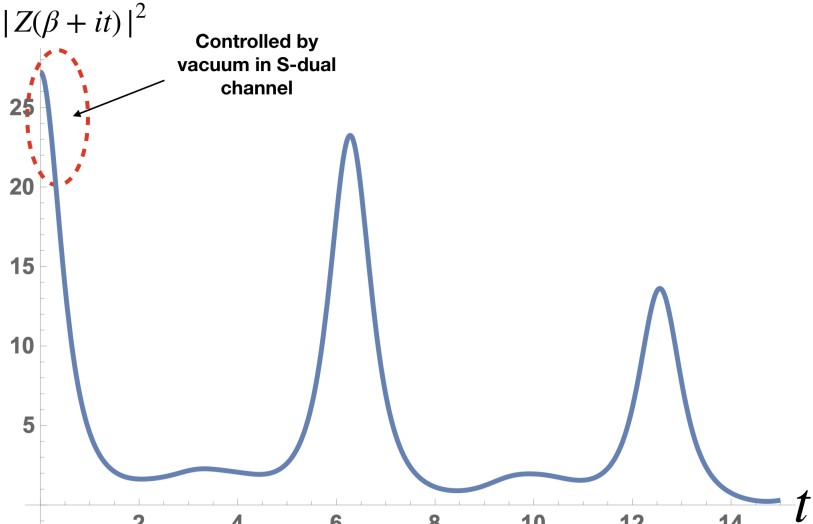

Figure 2: Spectral formfactor in 2d Ising ($\beta = 1$). It represents a typical behavior: it is large at early times and the initial decay is controlled by the vacuum in the S-dual channel. After a certain time a recurrence, generically only partial, happens. In chaotic theories the recurrence time is typically very long.

bound is similar)

$$\int_{\Delta-\delta}^{\Delta+\delta} d\Delta' \rho(\Delta') \leq e^{\beta\Delta} \left[ \int_{-\Lambda}^{\Lambda} dt \, Z_L\left(\frac{4\pi^2}{\beta+it}\right) \widehat{\phi}_+(t) e^{-itc/12} + \int_{-\Lambda}^{\Lambda} dt \, \left| Z_H\left(\frac{4\pi^2}{\beta+it}\right) \widehat{\phi}_+(t) \right| \right],$$
(15)

where we defined

$$Z_L(\beta) = \sum_{\Delta'<c/12} e^{-\beta(\Delta'-c/12)}, \qquad Z_H(\beta) = \sum_{\Delta'\geq c/12} e^{-\beta(\Delta'-c/12)}.$$
(16)

Now we show that the first term in (15) is dominated by $t = 0$, while the second term is dominated by $t = \Lambda$.

We estimate heavy operators by dropping the phases

$$\left| Z_H\left(\frac{4\pi^2}{\beta+it}\right) \right| \leq Z_H\left(\frac{4\pi^2\beta}{\beta^2+t^2}\right) \sim e^{\frac{t^2}{\beta}\frac{c}{12}}.$$
(17)

In the last estimate we again used modular invariance and that the integral over $t$ is dominated by $t \sim \Lambda \sim 1$, while $\beta \to 0$. Therefore, the effective inverse temperature is small $\frac{4\pi^2\beta}{\beta^2+t^2} \to 0$ and we can use the estimate by the dual vacuum. To justify the assumption $t \sim \Lambda \sim 1$ we notice that the contribution of an individual operator in $Z_H\left(\frac{4\pi^2\beta}{\beta^2+t^2}\right)$ is $\exp\left[-\left(\Delta'-\frac{c}{12}\right)\frac{4\pi^2\beta}{\beta^2+t^2}\right]$ with $\Delta' \geq c/12$. It is monotonically increasing with t. Therefore the integral is dominated by $t \sim \Lambda$.

In [25] it was said that to derive (17) one can use the Hartman-Keller-Stoica (HKS) bound [31]. In fact, as we just argued, only the high-temperature asymptotic of the partition function is needed.

Since $\widehat{\phi}_\pm(t)$ is continuous and has support $|t| < \Lambda$, we can estimate near $\Lambda$ that $\widehat{\phi}_\pm(t) = O(\Lambda - t)$ and therefore

$$\int_{-\Lambda}^{\Lambda} dt \, \left| Z_H\left(\frac{4\pi^2}{\beta+it}\right) \widehat{\phi}_+(t) \right| \sim \int_0^{\Lambda} dt \, (\Lambda-t) e^{\frac{t^2}{\beta}\frac{c}{12}} \sim \beta^2 e^{\frac{\Lambda^2}{\beta}\frac{c}{12}}.$$
(18)

The first integral in the RHS of (15) gets contributions from a finite number of light operators below $c/12$. It is dominated by the vacuum and gives

$$\int_{-\Lambda}^{\Lambda} dt\, Z_L\left(\frac{4\pi^2}{\beta+it}\right)\widehat{\phi}_+(t)e^{-itc/12} = \int_{-\Lambda}^{\Lambda} dt\, e^{\frac{4\pi^2}{\beta+it}\frac{c}{12}}\widehat{\phi}_+(t)e^{-itc/12} + \dots$$

$$= \sqrt{\frac{3}{\pi c}}\beta^{3/2}e^{\frac{4\pi^2}{\beta}\frac{c}{12}}\widehat{\phi}_+(0) + \dots \ . \tag{19}$$

Here the integral is dominated by the saddle $t = 0$ and the prefactor $\beta^{3/2}$ comes from integrating over fluctuations. Putting together the estimates we get from (15)

$$\int_{\Delta-\delta}^{\Delta+\delta} d\Delta'\rho(\Delta') \leq \sqrt{\frac{3}{\pi c}}\beta^{3/2}e^{\beta\Delta+\frac{4\pi^2}{\beta}\frac{c}{12}}\widehat{\phi}_+(0) + O\left(\beta^2 e^{\beta\Delta+\frac{\Lambda^2}{\beta}\frac{c}{12}}\right) . \tag{20}$$

Now it is clear that $\Lambda = 2\pi$ is the biggest value for which heavy operators are suppressed as we take $\beta \to 0$. From now on we will make this choice for $\Lambda$. Minimization of the first term in (20) over $\beta$ leads to the thermodynamic relation $\beta = \pi\sqrt{c/3\Delta}$.

Lower bound is similar. Finally, we have bounds at asymptotically large $\Delta$

$$2\pi\widehat{\phi}_-(0)\rho_0(\Delta) \leq \int_{\Delta-\delta}^{\Delta+\delta} d\Delta'\rho(\Delta') \leq 2\pi\widehat{\phi}_+(0)\rho_0(\Delta) , \tag{21}$$

$$\rho_0(\Delta) = \left(\frac{c}{48\Delta^3}\right)^{1/4}\exp\left(2\pi\sqrt{\frac{c\Delta}{3}}\right) . \tag{22}$$

## 2.1 Beurling-Selberg problem and Paley-Wiener theorem

At this point finding a bound on the density of states boils down to finding the functions $\phi_\pm(x)$ with the desired properties. Namely, we would like to solve the following problem.

**Beurling-Selberg problem:** Suppose $\phi_\pm(x)$ are continuous integrable functions with the following two properties:
  1) $\phi_-(x) \leq \theta_{[-\delta,\delta]}(x) \leq \phi_+(x)$, $\forall x \in \mathbb{R}$.
  2) Fourier $\widehat{\phi}_\pm(t)$ has finite support $|t| < \Lambda = 2\pi$.
Find the smallest value of $\widehat{\phi}_+(0)$ and the biggest value of $\widehat{\phi}_-(0)$.

Equivalently, we would like to minimize the area between $\phi_\pm$ and $\theta$ with the constraints that $\phi_\pm$ are bandlimited and bound $\theta$ from above/below.

This type of problem was first considered by Beurling [29] and Selberg [30] and has various applications in analytic number theory [32,33] and signal processing [34].

The Beurling-Selberg problem formulated above was solved for $2\delta \in \mathbb{Z}$ in [30] and for $2\delta \notin \mathbb{Z}$ in [34,35][7] Our task will be to simply use their results in the bounds (21). We describe the construction of [30] in section 3 and [35] in section 5. The following classic result will be very useful in solving the Beurling-Selberg problem.

**Theorem** (Paley-Wiener): Suppose $\phi \in L^2(\mathbb{R})$. Then $\phi$ can be extended to the complex plane as an entire function with $|\phi(z)| \leq Be^{\Lambda|z|}$ for some $B > 0$, if and only if Fourier $\widehat{\phi}(t)$ is supported on $|t| < \Lambda$.

---

[7]The reference [34] solved the problem when $0 < 2\delta < 1$. The reference [35] gave a complete solution for any $\delta$.

An entire function $\phi$ bounded by $|\phi(z)| < Be^{\Lambda|z|}$ in the complex plane is usually called a function of exponential type $\Lambda$. Therefore, Paley-Wiener theorem can be stated as equivalence between functions of exponential type $\Lambda$ and functions whose Fourier transform has finite support $[-\Lambda, \Lambda]$. The practical convenience of Paley-Wiener theorem is that one can determine the support of $\widehat{\phi}(t)$ simply by looking at the growth of $\phi(z)$ and vice versa.

For the full proof we refer the reader to, for example, [36]. It's easy to see that finite support of $\widehat{\phi}$ leads to the boundedness of $\phi$

$$|\phi(z)| = \left| \int_{-\Lambda}^{\Lambda} dt \, \widehat{\phi}(t) e^{itz} \right| \leq e^{\Lambda|z|} \int_{-\Lambda}^{\Lambda} dt \, |\widehat{\phi}(t)| = Be^{\Lambda|z|} \,. \tag{23}$$

The proof in the other direction proceeds in two steps. First, using Phragmén-Lindelöf principle one shows that boundedness condition of the theorem actually implies a stronger bound $|\phi(x+iy)| \leq Ce^{\Lambda|y|}$. Second, one can deform the contour parallel to the real line and estimate using the boundedness of $\phi$

$$|\widehat{\phi}(t)| = \left| \int \frac{dx}{2\pi} e^{ixt} \phi(x) \right| = \left| \int \frac{dx}{2\pi} e^{i(x+iy)t} \phi(x+iy) \right| \leq \# e^{-y(t-\Lambda)} \,. \tag{24}$$

Taking $y \to \infty$ we get $\widehat{\phi}(t) = 0$ for $t > \Lambda$. Similarly, shifting the contour to the lower half-plane gives $\widehat{\phi}(t) = 0$ for $t < -\Lambda$. See [36] for details or [37] for a more pedagogical discussion.

# 3 Extremal functions for $2\delta \in \mathbb{Z}$

In this section we assume $2\delta \in \mathbb{Z}$. The construction of [30] proceeds in two steps. First, one derives bounds on $\widehat{\phi}_{\pm}(0)$ from Poisson summation formula. Second, one constructs functions that saturate these bounds, thus showing their optimality.

## 3.1 A bound from Poisson summation

One easy way to get estimates on the allowed $\widehat{\phi}_{\pm}(0)$ is to use the Poisson resummation formula

$$2\pi \sum_{n \in \mathbb{Z}} e^{-2\pi i n r} \widehat{\phi}(2\pi n) = \sum_{n \in \mathbb{Z}} \phi(n+r) \,, \quad r \in [0,1) \,. \tag{25}$$

Applying this to $\phi_{\pm}$ and taking into account that $\widehat{\phi}_{\pm}(t)$ have support $|t| < 2\pi$, only one term in the LHS survives

$$2\pi \widehat{\phi}_{\pm}(0) = \sum_{n \in \mathbb{Z}} \phi_{\pm}(n+r) \,. \tag{26}$$

Since $\phi_{\pm}(x)$ bound $\theta_{[-\delta,\delta]}(x)$ from above and below, we can estimate the zero mode as

$$2\pi \widehat{\phi}_{+}(0) \geq \max_{r \in [0,1)} \sum_{n \in \mathbb{Z}} \theta_{[-\delta,\delta]}(n+r) \,, \tag{27}$$

$$2\pi \widehat{\phi}_{-}(0) \leq \min_{r \in [0,1)} \sum_{n \in \mathbb{Z}} \theta_{(-\delta,\delta)}(n+r) \,, \tag{28}$$

where we also optimized over $r \in [0,1)$ since $\widehat{\phi}_{\pm}(0)$ doesn't depend on it. The RHS in (27) is simply the maximum number of integer-spaced numbers $n+r$, that can be put into the interval $[-\delta, \delta]$. Similarly, the RHS of (28) is the minimum number of integer-spaced numbers $n+r$, that can be put into the interval $(-\delta, \delta)$.

While the bounds (27), (28) are true for any $\delta$, they are not always optimal. Later in this section we will construct functions $\phi_\pm$ that saturate (27), (28) when $2\delta \in \mathbb{Z}$. On the other hand, if $2\delta \notin \mathbb{Z}$ the bounds (27), (28) are not optimal and functions $\phi_\pm$ saturating them do not exist. We will discuss optimal bounds for $2\delta \notin \mathbb{Z}$ in section 5.

When $2\delta \in \mathbb{Z}$ the bounds (27), (28) are simply

$$2\pi\widehat{\phi}_+(0) \geq 2\delta + 1 , \qquad 2\delta \in \mathbb{Z} , \tag{29}$$

$$2\pi\widehat{\phi}_-(0) \leq 2\delta - 1 , \qquad 2\delta \in \mathbb{Z} . \tag{30}$$

To obtain these bounds we must choose $r$ in (27), (28) as follows. In the bound for $\widehat{\phi}_+$ it is clear that to attain the maximum number of integer-spaced numbers $n+r$ inside of the interval $[-\delta, \delta]$, we need to put one of the numbers $n+r$ at the edge of the interval, as should be clear from the figure 1. Therefore, if $\delta \in \mathbb{Z}$, then $n+r$ must be integers and $r = 0$. If $\delta \in \mathbb{Z} + \frac{1}{2}$, then $n+r$ must be half-integers and $r = \frac{1}{2}$. In the bound for $\widehat{\phi}_-$ we need to attain the minimum number of integer-spaced numbers inside of $(-\delta, \delta)$. The choice of $r$ is the same as for $\widehat{\phi}_+$, i.e. we need to put one of the numbers $n+r$ at the edge of the interval, see figure 1. To summarize, for $2\delta \in \mathbb{Z}$ we have two cases

$$\delta \in \mathbb{Z} , r = 0 , \quad \text{or} \quad \delta \in \mathbb{Z} + \frac{1}{2} , r = \frac{1}{2} . \tag{31}$$

## 3.2 Extremal functions

Now let's construct functions $\phi_\pm$ that saturate (29), (30). From the derivation of these bounds it's clear that saturation happens if

$$\phi_+(n+r) = 1 , \quad n+r \in [-\delta, \delta] , \tag{32}$$

$$\phi_+(n+r) = 0 , \quad n+r \notin [-\delta, \delta] , \tag{33}$$

$$\phi_+'(n+r) = 0 , \quad n+r \neq \pm\delta , \tag{34}$$

where the last condition on the derivative comes from the fact that $\phi_+(x) \geq \theta_{[-\delta,\delta]}(x)$ and therefore $\phi_+$ should be touching $\theta$ at the points $n+r$, see figure 1. Similarly, for $\phi_-$ we have

$$\phi_-(n+r) = 1 , \quad n+r \in (-\delta, \delta) , \tag{35}$$

$$\phi_-(n+r) = 0 , \quad n+r \notin (-\delta, \delta) , \tag{36}$$

$$\phi_-'(n+r) = 0 , \quad n+r \neq \pm\delta . \tag{37}$$

The functions $\phi_\pm(x)$ are essentially fixed by these properties. For $\delta \in \mathbb{Z}$ we consider

$$\phi_+(x) = \frac{\sin^2(\pi x)}{\pi^2}\left(\sum_{\substack{|n|\leq\delta \\ n\in\mathbb{Z}}} \frac{1}{(x-n)^2} + \frac{\lambda_+}{x+\delta} + \frac{\lambda_+}{\delta-x}\right) , \quad \delta \in \mathbb{Z} , \tag{38}$$

$$\phi_-(x) = \frac{\sin^2(\pi x)}{\pi^2}\left(\sum_{\substack{|n|<\delta \\ n\in\mathbb{Z}}} \frac{1}{(x-n)^2} + \frac{\lambda_-}{x+\delta} + \frac{\lambda_-}{\delta-x}\right) , \quad \delta \in \mathbb{Z} . \tag{39}$$

One can think of these functions as follows. Let's discuss $\phi_+$. We start with $\sin^2(\pi x)$, which vanishes at all integer $x$. These are the second order zeros that we want outside of $[-\delta, \delta]$. Then the role of the terms in parentheses in (38) is to cancel zeroes of $\sin^2(\pi x)$ inside of the interval $[-\delta, \delta]$. The residues of the 2nd order poles in parentheses are fixed by $\phi_+(n) = 1, |n| \leq \delta$. The first order poles are allowed only at the ends $x = \pm\delta$. This is

because we want $x = n, |n| < \delta$ to be minimums of $\phi_+$ in order for $\phi_+$ to bound $\theta$ from above. Finally, the residues of 1st order poles at $x = \pm\delta$ must be related in order to have $\phi_+(x) \sim \frac{1}{x^2}$, $x \to \infty$, so that $\phi_+$ is integrable. The constant term is not allowed for the same reason. Similar comments apply to $\phi_-$.

Note that the growth of (38), (39) in the complex plane is bounded by $|\phi_\pm(z)| \leq B e^{2\pi|z|}$. By Paley-Wiener theorem the Fourier transforms $\widehat{\phi}_\pm(t)$ have finite support $|t| < 2\pi$, just what we wanted our functions to satisfy.

Finally, we need to check that functions (38), (39) indeed bound the indicator function $\theta_{[-\delta,\delta]}(x)$. This imposes constraints on $\lambda_+, \lambda_-$. In appendix A we show that these constraints are given by

$$
\begin{aligned}
1 + \frac{1}{2\delta} \geq \lambda_+ \geq \delta\psi_1(\delta+1)\,, \\
1 - \frac{1}{2\delta} \leq \lambda_- \leq \delta\psi_1(\delta)\,,
\end{aligned}
\tag{40}
$$

where $\psi_1(z) = \frac{d^2}{dz^2} \log \Gamma(z)$ is the trigamma function.

Analogously, when $\delta \in \mathbb{Z} + \frac{1}{2}$ we consider

$$
\phi_+(x) = \frac{\cos^2(\pi x)}{\pi^2} \left( \sum_{\substack{|n|\leq\delta \\ n\in\mathbb{Z}+\frac{1}{2}}} \frac{1}{(x-n)^2} + \frac{\lambda_+}{x+\delta} + \frac{\lambda_+}{\delta-x} \right), \quad \delta \in \mathbb{Z} + \frac{1}{2}\,,
\tag{41}
$$

$$
\phi_-(x) = \frac{\cos^2(\pi x)}{\pi^2} \left( \sum_{\substack{|n|<\delta \\ n\in\mathbb{Z}+\frac{1}{2}}} \frac{1}{(x-n)^2} + \frac{\lambda_-}{x+\delta} + \frac{\lambda_-}{\delta-x} \right), \quad \delta \in \mathbb{Z} + \frac{1}{2}\,.
\tag{42}
$$

In this case $r = \frac{1}{2}$ (see (31)) and the zeros are at half-integers. These functions are fixed similarly to (38), (39). In particular, the constraints on $\lambda_\pm$ are the same (40).

By construction, the functions (38)-(42) saturate the bounds (29), (30). One can also check this directly by integrating. For both $\delta \in \mathbb{Z}$ and $\delta \in \mathbb{Z} + \frac{1}{2}$

$$
2\pi\widehat{\phi}_+(0) = \int_{-\infty}^{\infty} dx\, \phi_+(x) = 2\delta + 1\,,
\tag{43}
$$

$$
2\pi\widehat{\phi}_-(0) = \int_{-\infty}^{\infty} dx\, \phi_-(x) = 2\delta - 1\,.
\tag{44}
$$

Note that $\lambda_\pm$ - dependent terms integrate to zero. Inserting the constructed functions in (21), we find bounds on the number of states

$$
(2\delta - 1)\rho_0(\Delta) \leq \int_{\Delta-\delta}^{\Delta+\delta} d\Delta'\rho(\Delta') \leq (2\delta + 1)\rho_0(\Delta), \qquad 2\delta \in \mathbb{Z}\,.
\tag{45}
$$

## 3.3 Saturation at $c = 4, 8, 12, \ldots$

Selberg's functions (38)-(42) indeed give the best possible bounds for $2\delta \in \mathbb{Z}$ that can be obtained from (21). However, we are not guaranteed that there is a fully S-invariant partition function $Z(\beta)$ that saturates (45), since using bandlimited functions in (13) could be too crude in the first place. Here, we show that there is a zoo of S-invaraint partition functions for $c = 4k, k \in \mathbb{Z}_{>0}$ that saturate the bounds (45).

Let us consider two nice examples of $S$-invariant partition functions at $c = 4, 12$. They are given by ($q = e^{2\pi i \tau} = e^{-\beta}$)

$$Z_4(\beta) = j(\tau)^{1/3} = q^{-1/3}(1 + 248q + \dots), \tag{46}$$

$$Z_{12}(\beta) = j(\tau) - 744 = q^{-1}(1 + 196884q^2 + \dots), \tag{47}$$

where $j(\tau)$ is Klein's $j$-function. We consider (46), (47) as non-holomorphic partition functions with $\tau = -\bar\tau = i\frac{\beta}{2\pi}$. The condition $\tau = -\bar\tau$ explicitly breaks T-invariance $\tau \to \tau + 1, \bar\tau \to \bar\tau + 1$. Therefore, our discussion here concerns only S-invariant partition functions. They do not necessarily possess an extension to $SL(2, \mathbb{Z})$ invariant functions of $\tau, \bar\tau$. $SL(2, \mathbb{Z})$ invariant partition functions, constructed by combining holomorphic and anti-holomorphic parts, will be considered in section 6.

The dimensions of operators in both partition functions are given by non-negative integers $\Delta_k = k \in \mathbb{Z}_{\geq 0}$. The degeneracy $d_k$ of an operator $\Delta_k$ at large $k$ is[8]

$$d_k = \rho_0(k) + \dots, \tag{48}$$

where $\rho_0$ is defined in (22) and corrections are suppressed at $k \to \infty$.

Now it is easy to check that the bounds (45) are saturated. First, recall that the upper bound always bounds the number of operators in the window $[\Delta - \delta, \Delta + \delta]$ where we include the "edge states" at $\Delta \pm \delta$. While the lower bound holds for the number of operators in $(\Delta - \delta, \Delta + \delta)$ where we do not include the edge states. See the discussion below (11).

For the upper bound, we are counting the number of states in the interval $[\Delta - \delta, \Delta + \delta]$ of size $2\delta \in \mathbb{Z}$. There can be at most $2\delta + 1$ integers in this interval, each corresponding to an operator with degeneracy $\rho_0(\Delta)$. Thus we have $(2\delta + 1)\rho_0(\Delta)$ operators.

For the lower bound we are counting states in the interval $(\Delta - \delta, \Delta + \delta)$. There are at least $2\delta - 1$ integers in this interval, each corresponding to a state with degeneracy $\rho_0(\Delta)$, thus giving $(2\delta - 1)\rho_0(\Delta)$ states in total.

More generally, the bounds (13) would be saturated[9] if the functions $\phi_\pm$ take the following values on the physical spectrum $\Delta_{ph}$

$$\phi_+(\Delta_{ph}) = \begin{cases} 0, & \Delta_{ph} \notin [\Delta - \delta, \Delta + \delta], \\ 1, & \Delta_{ph} \in [\Delta - \delta, \Delta + \delta], \end{cases}$$
$$\phi_-(\Delta_{ph}) = \begin{cases} 0, & \Delta_{ph} \notin (\Delta - \delta, \Delta + \delta), \\ 1, & \Delta_{ph} \in (\Delta - \delta, \Delta + \delta). \end{cases} \tag{49}$$

This is indeed the case for $c = 4, 12$ partition functions (46), (47) and Selberg's functions (38)-(42). The Selberg's functions vanish at the physical spectrum $\Delta_{ph} = k \in \mathbb{Z}$ outside and are one inside of the corresponding interval.

With the understanding of (49) it is clear that any partition function with an integer-spaced spectrum saturates the bounds (45), because Selberg's functions (38)-(42) satisfy (49) in this case. A large class of such partition functions is given by

$$Z_{4a+12k}(\beta) = j(\tau)^{a/3} P_k(j(\tau)), \qquad a = 0, 1, 2 \, ; k \in \mathbb{Z}_{\geq 0}, \tag{50}$$

where $P_k(x) = x^k + \dots$ is a monic (to ensure that the vacuum is unique) polynomial such that $q$-expansion coefficients of the partition function are non-negative. These partition functions correspond to the central charges $c = 4a + 12k = 4, 8, 12, 16, \dots$ and were previously considered in [42].

---

[8]This follows from the classic results of Rademacher and Zuckerman [38–40]. One can also derive this using Ingham's theorem [41], which we explain in the appendix B.

[9]We also lose precision when we multiply (11) by $e^{\beta(\Delta - \Delta' \pm \delta)}$. However, this factor is unimportant at large $\Delta$, where we take $\beta = \pi\sqrt{\frac{c}{3\Delta}} \to 0$ and $\Delta' \approx \Delta$.

The functions $\phi_\pm(\Delta)$ are reminiscent of the conformal bootstrap extremal functionals (e.g. [43–49]): outside of a certain interval they are non-negative/non-positive and vanish at the dimensions of physical operators.

# 4   A simple non-optimal bound for $2\delta \notin \mathbb{Z}$

In this section we will show that the bounds (45) in fact hold for any $\delta \geq 0$. They turn out to be sub-optimal when $2\delta \notin \mathbb{Z}$ in the sense that they do not minimize/maximize $\widehat{\phi}_\pm(0)$ with the constraints described in section 2.1. We will discuss the optimal functions for $2\delta \notin \mathbb{Z}$ in the next section. However, the optimal functions in the general case are more complicated and we would like to start with a simpler sub-optimal bound.

First, note that it is not completely trivial to generalize (38),(39),(41),(42) to non-integer $2\delta$ because of the sum in parenthesis. However, one can use the following trick due to Selberg [30]. Let's consider the upper bound. We would like to bound

$$\theta_{[-\delta,\delta]}(x) = \frac{1}{2}\mathrm{sgn}(\delta + x) + \frac{1}{2}\mathrm{sgn}(\delta - x) \tag{51}$$

from above. Therefore, we can first take the function (38) and construct a function that bounds $\mathrm{sgn}(x)$ as

$$B_+(x) = \lim_{\substack{\delta \to \infty \\ \delta \in \mathbb{Z}}} \left( 2\phi_+(x - \delta) - 1 \right) \tag{52}$$

$$= \frac{2\sin^2(\pi x)}{\pi^2} \left[ \sum_{k=0}^{\infty} \frac{1}{(x-k)^2} + \frac{1}{x} \right] - 1 \geq \mathrm{sgn}(x) \,. \tag{53}$$

Note that the only admissible choice (40) as $\delta \to \infty$ is $\lambda_\pm = 1$. Now we consider a new function $\phi_+$ that bounds $\theta$ for any $\delta \geq 0$

$$\phi_+(x) = \frac{1}{2}B_+(\delta + x) + \frac{1}{2}B_+(\delta - x) \geq \theta_{[-\delta,\delta]}(x) \,. \tag{54}$$

Similarly, for the lower bound

$$B_-(x) = \lim_{\substack{\delta \to \infty \\ \delta \in \mathbb{Z}}} \left( 2\phi_-(x - \delta) - 1 \right) \tag{55}$$

$$= \frac{2\sin^2(\pi x)}{\pi^2} \left[ \sum_{k=1}^{\infty} \frac{1}{(x-k)^2} + \frac{1}{x} \right] - 1 \leq \mathrm{sgn}(x) \,, \tag{56}$$

$$\phi_-(x) = \frac{1}{2}B_-(\delta + x) + \frac{1}{2}B_-(\delta - x) \leq \theta_{(-\delta,\delta)}(x) \,. \tag{57}$$

The functions $B_\pm(x)$ were first considered by Beurling [29]. One can show that the functions (54), (57) give the same result (43), (44) for the zero mode $\widehat{\phi}_\pm(0)$ and also reduce to (38),(39),(41),(42) for $2\delta \in \mathbb{Z}$. See appendix C for details. Thus, we have a bound for any $\delta$

$$(2\delta - 1)\rho_0(\Delta) \leq \int_{\Delta-\delta}^{\Delta+\delta} d\Delta' \rho(\Delta') \leq (2\delta + 1)\rho_0(\Delta), \qquad \delta \geq 0 \,. \tag{58}$$

Two comments are in order. First, the lower bound shows that in any window of size $2\delta > 1$ there is a non-zero number of operators, the result previously established in [25, 26]. Second, as we take $\delta \to 0$ the upper bound becomes $\rho_0(\Delta)$. This implies that the maximum degeneracy of an individual operator is at most $\rho_0(\Delta)$ up to additive error terms suppressed at asymptotically large $\Delta$. In fact, partition functions $c = 4, 12, 16, \ldots$ considered in section 3.3 saturate this bound on degeneracy, see section 3.3.

# 5 Extremal functions for $2\delta \notin \mathbb{Z}$

When the size of the window is not integer $2\delta \notin \mathbb{Z}$ the functions (54), (57) we constructed previously turn out not to be optimal. The optimal functions in non-integer case were found by Littmann in [35]. In this section we describe his results and their implications for our bounds.

## 5.1 Generalized Poisson summation

In the integer case it was very useful, for both deriving functions $\phi_{\pm}$ and proving their optimality, to use Poisson summation formula (26) for functions with finite Fourier support. A generalization that is useful in non-integer case was found by Littman [35]. Suppose a continuous integrable function $\phi(x)$ has Fourier of support[10] $|t| < 2\pi$. Then

$$2\pi\widehat{\phi}(0) = \sum_{x_n} \phi(x_n) \frac{\pi(x_n^2 + \gamma^2)}{\gamma + \pi(x_n^2 + \gamma^2)} \,, \tag{59}$$

where $\gamma > 0, r \in \mathbb{R}$ and $x_n, n \in \mathbb{Z}$ are the roots $B(x_n) = 0$ of

$$B(x) = x \sin \pi(x + r) - \gamma \cos \pi(x + r) \,. \tag{60}$$

In particular, if we take $\gamma \to \infty$ in (59), the roots become integer spaced $x_n = n + \tilde{r}$ and we recover the Poisson formula (26). But (59) is true for any $\gamma > 0, r \in \mathbb{R}$.[11]

To bound $\theta_{[-\delta,\delta]}(x)$ we would like $\pm\delta$ to be among the nodes $x_n$, similarly to the integer case. This determines $\gamma, r$ that we take. The analytic expressions for $\gamma, r$ depend on the fractional part $\{\delta\} = \delta - [\delta]$

$$B(\pm\delta) = 0 \,, \tag{61}$$

$$\Rightarrow \quad 0 < \{\delta\} < \frac{1}{2}: \quad \gamma = \delta \tan(\pi\delta) > 0 \,, \; r = 0 \,, \tag{62}$$

$$\frac{1}{2} < \{\delta\} < 1: \quad \gamma = -\delta \cot(\pi\delta) > 0 \,, \; r = \frac{1}{2} \,. \tag{63}$$

The function $B(x)$ then takes the form

$$B(x) = \begin{cases} x \sin(\pi x) - \delta \tan(\pi\delta) \cos(\pi x) \,, & 0 < \{\delta\} < \frac{1}{2} \\ x \cos(\pi x) - \delta \cot(\pi\delta) \sin(\pi x) \,, & \frac{1}{2} < \{\delta\} < 1 \,. \end{cases} \tag{64}$$

Similarly to (27, 28), applying (59) to $\phi_{\pm}$ we find

$$2\pi\widehat{\phi}_+(0) \geq \sum_{|x_n| \leq \delta} \frac{\pi(x_n^2 + \gamma^2)}{\gamma + \pi(x_n^2 + \gamma^2)} \,, \tag{65}$$

$$2\pi\widehat{\phi}_-(0) \leq \sum_{|x_n| < \delta} \frac{\pi(x_n^2 + \gamma^2)}{\gamma + \pi(x_n^2 + \gamma^2)} \,. \tag{66}$$

These bounds can be saturated and we construct the corresponding functions $\phi_{\pm}$ next.

---

[10]Both (26) and (59) can be easily generalized to functions of support $|t| < \Lambda$ by a rescaling of variables.

[11]Littmann's formula (59) has a beautiful interpretation in terms of de Branges reproducing kernel Hilbert spaces of entire function [50]. Roughly speaking, if $\phi(z) = f(z)\overline{f(\bar{z})}$, where $\phi$ is of exponential type $2\pi$ (i.e. Fourier of support $2\pi$) and $f$ is of exponential type $\pi$, then the LHS of the formula (59) can be interpreted as the norm of $f$ in a certain Hilbert space. And the RHS is the same norm written as an expansion over an orthonormal basis in this Hilbert space. We prefer not to delve into details about this interpretation, as this would take us too far from our goals. We refer the interested reader to [35].

## 5.2 Extremal functions

The idea of the construction is analogous to section 3. We start with $\phi_+$. To saturate (65) we must have

$$\phi_+(x_n) = 0, \quad |x_n| > \delta , \tag{67}$$

$$\phi_+(x_n) = 1, \quad |x_n| \leq \delta , \tag{68}$$

$$\phi'_+(x_n) = 0, \quad x_n \neq \pm\delta . \tag{69}$$

The last condition on the derivative comes from the fact that $\phi_+(x) \geq \theta_{[-\delta,\delta]}(x)$, so the nodes $x_n \neq \pm\delta$ must be local minimums. We take the following ansatz

$$\phi_+(x) = B(x)^2 \sum_{|x_n| \leq \delta} \left( \frac{a(x_n)}{(x-x_n)^2} + \frac{b(x_n)}{x-x_n} \right) . \tag{70}$$

Note that by Paley-Wiener theorem the Fourier transform $\widehat{\phi}(t)$ has support $|t| < 2\pi$. The factor $B(x)^2$ ensures that (67, 69) are satisfied for $|x_n| > \delta$, i.e. we have 2nd order zeros outside of the interval. A simple calculation shows that the conditions (68, 69) for $|x_n| \leq \delta$ determine almost all $a(x_n), b(x_n)$

$$a(x_n) = \frac{1}{B'(x_n)^2} = \frac{x_n^2 + \gamma^2}{[\gamma + \pi(x_n^2 + \gamma^2)]^2} , \tag{71}$$

$$b(x_n) = -\frac{B''(x_n)}{B'(x_n)^3} = -\frac{2\pi x_n(x_n^2 + \gamma^2)}{[\gamma + \pi(x_n^2 + \gamma^2)]^3}, \quad x_n \neq \pm\delta . \tag{72}$$

Now the only undetermined coefficients are $b(\pm\delta)$. We fix them by requiring that $\phi_+$ is integrable

$$\phi_+(x) \sim \frac{1}{x^2}, \quad x \to \infty . \tag{73}$$

The cancellation of order $x, 1, \frac{1}{x}$ terms in $x \to \infty$ expansion leads to

$$\sum_{|x_n| \leq \delta} b(x_n) = 0 , \tag{74}$$

$$\sum_{|x_n| \leq \delta} (x_n b(x_n) + a(x_n)) = 0 , \tag{75}$$

$$\sum_{|x_n| \leq \delta} \left( x_n^2 b(x_n) + 2x_n a(x_n) \right) = 0 . \tag{76}$$

To solve these constraints, note that if $x_n$ is a root of $B(x)$, then $-x_n$ is also a root, see (64). Then (71, 72) imply $a(x_n) = a(-x_n)$ and $b(x_n) = -b(-x_n), x_n \neq \pm\delta$. Now the constraint (74) requires that the same symmetry is true for $x_n = \pm\delta$, i.e. $b(\delta) = -b(-\delta)$. Due to these symmetries of $a(x_n), b(x_n)$ the constraint (76) is automatically satisfied. The remaining constraint (75) determines $b(\delta)$

$$b(\delta) = -\frac{\delta^2 + \gamma^2}{\delta[\gamma + \pi(\delta^2 + \gamma^2)]^2} - \frac{1}{2\delta} \sum_{|x_n| < \delta} \frac{(x_n^2 + \gamma^2)[\gamma - \pi(x_n^2 - \gamma^2)]}{[\gamma + \pi(x_n^2 + \gamma^2)]^3} . \tag{77}$$

The function $\phi_-(x)$ is constructed in a similar manner. We require

$$\phi_-(x_n) = 0, \quad |x_n| \geq \delta , \tag{78}$$

$$\phi_-(x_n) = 1, \quad |x_n| < \delta , \tag{79}$$

$$\phi'_-(x_n) = 0, \quad x_n \neq \pm\delta , \tag{80}$$

and we find

$$\phi_-(x) = B(x)^2 \left[ \sum_{|x_n|<\delta} \left( \frac{a(x_n)}{(x-x_n)^2} + \frac{b(x_n)}{x-x_n} \right) + \frac{c(\delta)}{x-\delta} + \frac{c(-\delta)}{x+\delta} \right] . \tag{81}$$

The analysis at $x_n \neq \pm\delta$ is the same and therefore the coefficients $a(x_n), b(x_n)$ are given by (71, 72). Requiring $\phi_-(x) \sim \frac{1}{x^2}, x \to \infty$ leads to

$$c(\delta) = -c(-\delta) = -\frac{1}{2\delta} \sum_{|x_n|<\delta} \frac{(x_n^2+\gamma^2)[\gamma - \pi(x_n^2-\gamma^2)]}{[\gamma + \pi(x_n^2+\gamma^2)]^3} . \tag{82}$$

Finally, one needs to check that $\phi_\pm(x)$ in (70, 81) indeed bound $\theta_{[-\delta,\delta]}(x)$ from above and below. This was proved in [35].[12]

To summarize, we constructed optimal functions (70, 81) that saturate (65,66)

$$2\pi\widehat{\phi}_+(0) = \sum_{|x_n|\leq\delta} \frac{\pi(x_n^2+\gamma^2)}{\gamma + \pi(x_n^2+\gamma^2)} , \tag{83}$$

$$2\pi\widehat{\phi}_-(0) = \sum_{|x_n|<\delta} \frac{\pi(x_n^2+\gamma^2)}{\gamma + \pi(x_n^2+\gamma^2)} , \tag{84}$$

where $x_n$ are roots of $B(x)$ defined in (60) and $\gamma, r$ are given by (62, 63). In general, the nodes $x_n$ are some transcendental numbers that we can't write down explicitly. However, the difference $\widehat{\phi}_+(0) - \widehat{\phi}_-(0)$ gets a contribution only from $x_n = \pm\delta$ and takes a simple form

$$2\pi(\widehat{\phi}_+(0) - \widehat{\phi}_-(0)) = \frac{2}{1 + \left| \frac{\sin(2\pi\delta)}{2\pi\delta} \right|} \leq 2 . \tag{85}$$

This already demonstrates that (83, 84) give better bounds than (58), where the difference was 2.

When $\delta$ is sufficiently small, the only roots in the range $|x_n| \leq \delta$ are $x_n = 0, \pm\delta$ and the expressions (83, 84) take a simple form. Namely

$$0 < 2\delta < 1: \quad 2\pi\widehat{\phi}_+(0) = \frac{2}{1 + \frac{\sin(2\pi\delta)}{2\pi\delta}} , \quad \widehat{\phi}_-(0) = 0 , \tag{86}$$

$$1 < 2\delta < 2: \quad 2\pi\widehat{\phi}_+(0) = \frac{2}{1 - \frac{\sin(2\pi\delta)}{2\pi\delta}} + \frac{1}{1 - \frac{\tan(\pi\delta)}{\pi\delta}} , \tag{87}$$

$$2\pi\widehat{\phi}_-(0) = \frac{1}{1 - \frac{\tan(\pi\delta)}{\pi\delta}} . \tag{88}$$

For larger $\delta$ one can find roots of (60) numerically. We plot (83,84) as functions of $\delta$ in the figure 3.

## 6 Fixed spin

Now we generalize the discussion of previous sections to operators of fixed spin $J$. We turn on an angular potential $\Omega$ and consider the grand canonical partition function

$$Z(\beta, \Omega) = \sum_{h, \bar{h}} e^{-\beta\left(h+\bar{h}-\frac{c}{12}\right)} e^{-i\Omega(h-\bar{h})} . \tag{89}$$

---

[12]The proof of this fact in [35] was given for a much more general choice of the functions $\phi_\pm(x)$. We expect that it can be considerably simplified for the particular functions we study and one should be able to avoid the subtleties of the general case, though we haven't done it.

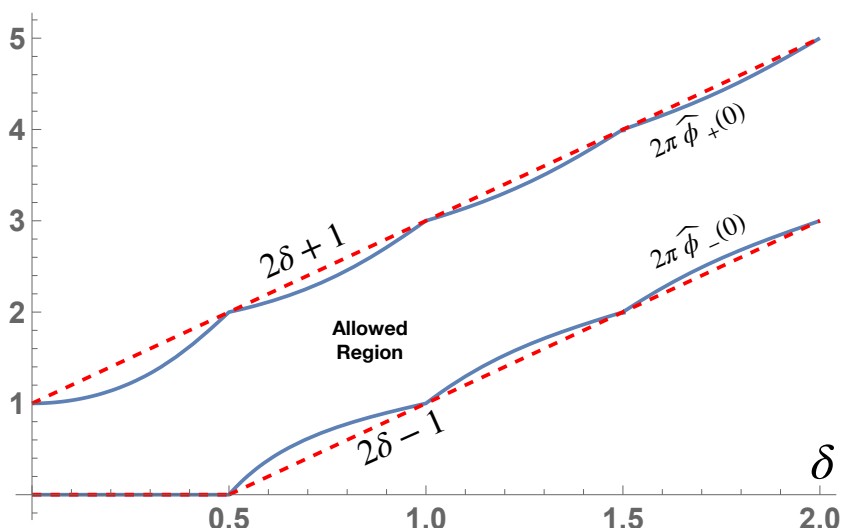

Figure 3: The solid blue lines are the optimal values (83,84) of $2\pi\widehat{\phi}_\pm(0)$ as functions of $\delta$. The number of operators in a window of size $2\delta$ is bounded by $2\pi\widehat{\phi}_-(0)\rho_0(\Delta)\leq\int_{\Delta-\delta}^{\Delta+\delta}d\Delta'\rho(\Delta')\leq 2\pi\widehat{\phi}_+(0)\rho_0(\Delta)$. The dashed red lines are from the bounds (58), which are optimal only for $2\delta\in\mathbb{Z}$. The blue and red lines touch when $2\delta\in\mathbb{Z}$.

The dimension $\Delta$ and spin $J$ are given by $h+\bar{h}$ and $|h-\bar{h}|\in\mathbb{Z}$ respectively. The main technical difference with the previous discussion will be the double sum over $h,\bar{h}$ instead of a single sum over $\Delta$. This will lead to a different splitting to light and heavy operators and to a different value $\Lambda$ of the support of $\widehat{\phi}_\pm(t)$.

First, we project onto operators of fixed spin $J$

$$Z_J(\beta)=\int_{-\pi}^{\pi}\frac{\mathrm{d}\Omega}{2\pi}\left(\frac{e^{i\Omega J}+e^{-i\Omega J}}{1+\delta_{J,0}}\right)Z(\beta,\Omega)=\int_0^\infty d\Delta\,\rho_J(\Delta)e^{-\beta(\Delta-c/12)}. \tag{90}$$

Our goal is to derive bounds for the density of states $\rho_J(\Delta)$ at large $\Delta$ and fixed spin $J$. The analog of (13) is

$$e^{\beta(\Delta-\delta-c/12)}\int_{-\infty}^{\infty}dt\,Z_J(\beta+it)\widehat{f}_-(t)e^{-itc/12}\leq\int_{\Delta-\delta}^{\Delta+\delta}d\Delta'\rho_J(\Delta') \tag{91}$$

$$\leq e^{\beta(\Delta+\delta-c/12)}\int_{-\infty}^{\infty}dt\,Z_J(\beta+it)\widehat{f}_+(t)e^{-itc/12}\,,$$

where $f_\pm(x)$ are functions that bound $\theta_{[-\delta,\delta]}(x)$ from above/below and have Fourier $\widehat{f}_\pm(t)$ of support $|t|<\Lambda$. We changed notation for these functions to distinguish from the previous sections. To use (91) we would like to estimate the integrals

$$\int_{-\Lambda}^{\Lambda}dt\int_{-\pi}^{\pi}\frac{\mathrm{d}\Omega}{2\pi}\cos(\Omega J)Z(\beta+it,\Omega)\widehat{f}_\pm(t)e^{-itc/12}\,. \tag{92}$$

To do that we go to the dual channel $\tau \to -1/\tau, \bar\tau \to -1/\bar\tau$

$$\tau = \frac{1}{2\pi}\left(i(\beta + it) + \Omega\right), \quad \bar\tau = \frac{1}{2\pi}\left(-i(\beta + it) + \Omega\right), \tag{93}$$

$$-\frac{1}{\tau} = \frac{1}{2\pi}\left(i\frac{4\pi^2\beta}{\beta^2 + (\Omega - t)^2} - \frac{4\pi^2(\Omega - t)}{\beta^2 + (\Omega - t)^2}\right), \tag{94}$$

$$-\frac{1}{\bar\tau} = \frac{1}{2\pi}\left(-i\frac{4\pi^2\beta}{\beta^2 + (\Omega + t)^2} - \frac{4\pi^2(\Omega + t)}{\beta^2 + (\Omega + t)^2}\right). \tag{95}$$

and split into light and heavy operators

$$\int_{-\Lambda}^{\Lambda} dt \int_{-\pi}^{\pi} \frac{d\Omega}{2\pi} \cos(\Omega J)\left(Z'_L(\beta + it, \Omega) + Z'_H(\beta + it, \Omega)\right)\widehat{f}_\pm(t) e^{-itc/12}, \tag{96}$$

where primes denote the dual channel (94), (95). Light (L) and heavy (H) operators are defined by[13]

$$\text{Light:} \quad h < \frac{c}{24} \quad \text{and} \quad \bar h < \frac{c}{24}, \tag{97}$$

$$\text{Heavy:} \quad h \geq \frac{c}{24} \quad \text{or} \quad \bar h \geq \frac{c}{24}. \tag{98}$$

The convenience of this splitting is that there are only finite number of light operators and thus the first term in (96) will be dominated by the vacuum in the limit $\beta \to 0$.

In the light sector the integral is dominated by $t = 0$ and $\Omega = 0$. Using saddle-point approximation we have

$$\int_{-\Lambda}^{\Lambda} dt \int_{-\pi}^{\pi} \frac{d\Omega}{2\pi} \cos(\Omega J) Z'_L(\beta + it, \Omega)\widehat{f}_\pm(t) e^{-itc/12}$$
$$\simeq \int_{-\Lambda}^{\Lambda} dt \int_{-\pi}^{\pi} \frac{d\Omega}{2\pi} \cos(\Omega J) \exp\left(\frac{4\pi^2(\beta + it)}{(\beta + it)^2 + \Omega^2}\frac{c}{12}\right)\widehat{f}_\pm(t) e^{-itc/12} \tag{99}$$
$$\simeq \frac{3}{2\pi^2 c}\widehat{f}_\pm(0)\,\beta^3 e^{\frac{4\pi^2}{\beta}\frac{c}{12}} + \dots .$$

For the heavy part, dropping the phases, i.e. retaining only the first term in each of (94), (95), we can estimate

$$\left|Z'_H(\beta + it, \Omega)\right| \leq \sum_{\substack{h, \bar h \\ \max(h, \bar h) \geq c/24}} \exp\left[-\frac{4\pi^2\beta}{\beta^2 + (\Omega - t)^2}(h - c/24) - \frac{4\pi^2\beta}{\beta^2 + (\Omega + t)^2}(\bar h - c/24)\right]. \tag{100}$$

First, we estimate as follows

$$\left|Z'_H(\beta + it, \Omega)\right| \leq e^{\frac{4\pi^2}{\beta}\frac{c}{24}} \sum_{\substack{h \geq \bar h \\ h \geq c/24}} \exp\left[-\frac{4\pi^2\beta}{\beta^2 + (\Omega - t)^2}(h - c/24) - \frac{4\pi^2\beta}{\beta^2 + (\Omega + t)^2}\bar h\right]$$
$$+ e^{\frac{4\pi^2}{\beta}\frac{c}{24}} \sum_{\substack{h < \bar h \\ \bar h \geq c/24}} \exp\left[-\frac{4\pi^2\beta}{\beta^2 + (\Omega - t)^2}h - \frac{4\pi^2\beta}{\beta^2 + (\Omega + t)^2}(\bar h - c/24)\right]. \tag{101}$$

Here the expressions in the exponentials under the sums are always positive. Therefore, the integrals over $t, \Omega$ are dominated by the regions, where the effective inverse temperatures

---

[13]Note that our definition of heavy/light operators is different from [31].

$\beta_\pm = \frac{4\pi^2\beta}{\beta^2+(\Omega\pm t)^2}$ are the smallest. In particular, as we will verify below, there are such $\Omega, t$, that $\beta_\pm \to 0$. In this case we estimate (101) by the vacuum in the S-dual channel

$$\left| Z'_H(\beta + it, \Omega) \right| \lesssim \exp\left( \frac{4\pi^2}{\beta} \frac{c}{24} + \frac{(\Omega - t)^2}{\beta} \frac{c}{24} + \frac{(\Omega + t)^2}{\beta} \frac{c}{24} \right). \tag{102}$$

Using this estimate we have

$$\left| \int_{-\Lambda}^{\Lambda} dt \int_{-\pi}^{\pi} \frac{d\Omega}{2\pi} \cos(\Omega J) Z'_H(\beta + it, \Omega) \widehat{f}_\pm(t) e^{-itc/12} \right|$$
$$\lesssim e^{\frac{4\pi^2}{\beta} \frac{c}{24}} \int_{-\Lambda}^{\Lambda} dt \int_{-\pi}^{\pi} \frac{d\Omega}{2\pi} \exp\left( \frac{\Omega^2 + t^2}{\beta} \frac{c}{12} \right) \widehat{f}_\pm(t) \tag{103}$$
$$\sim \beta^3 \exp\left( \frac{4\pi^2}{\beta} \frac{c}{24} + \frac{\pi^2 + \Lambda^2}{\beta} \frac{c}{12} \right),$$

where we also used that $\widehat{f}_\pm(t) \sim \Lambda \mp t$ is vanishing near the end of the support $\pm\Lambda$. The integrals were computed by expanding near $t = \pm\Lambda, \Omega = \pm\pi$.

Collecting the results we have, for example, for the upper bound

$$\int_{\Delta-\delta}^{\Delta+\delta} d\Delta' \rho_J(\Delta') \leq \frac{1}{1+\delta_{J,0}} \frac{3}{\pi^2 c} \widehat{f}_+(0) \beta^3 e^{\beta\Delta + \frac{4\pi^2}{\beta} \frac{c}{12}} + O\left( \beta^3 e^{\beta\Delta + \frac{3\pi^2 + \Lambda^2}{\beta} \frac{c}{12}} \right). \tag{104}$$

For heavy operators to be suppressed in $\beta \to 0$ limit, we set $\Lambda = \pi - \beta^{1/4}$. Note, that for this $\Lambda$ and $\Omega = \pi, t = \Lambda$ the effective inverse temperatures in (101) are small $\beta_\pm \to 0$. Namely, $\frac{4\pi^2\beta}{\beta^2+(\pi-\Lambda)^2} \sim \sqrt{\beta} \to 0$. So that the approximation (102) is justified.[14] Similar comments apply to the lower bound.

As before, we set $\beta = \pi\sqrt{c/3\Delta}$. In the $\Delta \to \infty$ limit, we obtain bounds

$$\pi\widehat{f}_-(0)\rho_J^0(\Delta) \leq \int_{\Delta-\delta}^{\Delta+\delta} d\Delta' \rho_J(\Delta') \leq \pi\widehat{f}_+(0)\rho_J^0(\Delta), \tag{105}$$

where $\rho_J^0(\Delta)$ is given by

$$\rho_J^0(\Delta) \equiv \frac{2}{1+\delta_{J,0}} \sqrt{\frac{c}{12\Delta^3}} \exp\left[ 2\pi\sqrt{\frac{c\Delta}{3}} \right]. \tag{106}$$

Now we turn our attention to finding optimal $\widehat{f}_\pm(0)$. The results obtained in the previous sections are almost readily applicable upon proper scaling. We go through the generalized version of the salient equations appearing in § 3, 4, 5 below. The only difference is that we need functions with Fourier of support $\Lambda = \pi - \beta^{1/4}$ instead of $2\pi$. This is achieved by considering

$$f_\pm(x) = \phi_\pm\left( \frac{\Lambda}{2\pi} x \right)\Big|_{\delta \to \frac{\Lambda}{2\pi}\delta}, \tag{107}$$

$$\widehat{f}_\pm(t) = \frac{2\pi}{\Lambda} \widehat{\phi}_\pm\left( \frac{2\pi}{\Lambda} t \right)\Big|_{\delta \to \frac{\Lambda}{2\pi}\delta}, \tag{108}$$

---

[14]More generally, we could take $\Lambda = \pi - \beta^\alpha, 0 < \alpha < 1/2$. Heavy operators are suppressed and the approximation (102) is justified for this choice.

where $\phi_{\pm}(x)$ are functions with Fourier of support $2\pi$ considered in § 3, 4, 5. The rescaling of $\delta$ is needed so that $f_{\pm}(x)$ bound $\theta_{[-\delta,\delta]}(x)$ from above/below.

- Beurling-Selberg § 3, 4 : Taking functions (54),(57) that satisfy (43),(44) and setting $\Lambda = \pi$ in (108) we obtain

$$\pi\widehat{f}_{\pm}(0) = \delta \pm 1 \,. \tag{109}$$

Therefore, the bounds (105) become

$$(\delta - 1)\rho_J^0(\Delta) \leq \int_{\Delta-\delta}^{\Delta+\delta} d\Delta'\rho_J(\Delta') \leq (\delta + 1)\rho_J^0(\Delta)\,. \tag{110}$$

When $2\delta \cdot \frac{\Lambda}{2\pi} = \delta \in \mathbb{Z}$, these bounds are optimal among those that can be obtained with bandlimited functions $f_{\pm}$. They are saturated if the spacing between operators of spin $J$ is $\Delta_{n+1} - \Delta_n = 2$. We can construct such examples from the partition functions considered in § 3.3. We consider the partition functions (50) as $c_L = 8k$ chiral partition functions. We tensor each of them with their anti-chiral copies to obtain a partition function with $c_L = c_R = 8k$. These have integer spaced spectrum $h, \bar{h} \in \mathbb{Z}$. For a fixed $J = |h - \bar{h}|$, the spacing between operators $\Delta = J + 2\min(h, \bar{h})$ is 2 and the asymptotic degeneracy is exactly given by $\rho_J^0(\Delta)$, hence these partition functions saturate the bound. The factors $\delta \pm 1$ come from counting how many numbers spaced by 2 can be fit in the interval of length $2\delta \in 2\mathbb{Z}$.

A few simple consequences of (110) are:

1. The lower bound implies that in any window of size $2\delta > 2$ there is a non-zero number of operators of given spin $J$. This is again not impressive if the theory has Virasoro symmetry. However, the analysis can be easily generalized to counting Virasoro primaries for $c > 1$ with the same result.

2. The upper bound, in the $\delta \to 0$ limit, implies that the maximum degeneracy of an individual operator with dimension $\Delta$ and spin $J$ is $\rho_J^0(\Delta)$.

3. In any unitary 2d CFT operators of all spins $J \in \mathbb{Z}$ must be present. Again, this is trivial if Virasoro symmetry is present. However, the analysis can be easily generalized to counting Virasoro primaries for $c > 1$ with the same result.

- Littmann § 5: When $\delta \notin \mathbb{Z}$ the bounds (110) are not optimal. The optimal bounds are obtained from the functions (70), (81). Taking the equations (83), (84) and setting $\Lambda = \pi$ in (108), we get for the zero modes

$$\pi\widehat{f}_+(0) = \sum_{|x_n|\leq\delta} \frac{\pi(\frac{x_n^2}{4} + \hat{\gamma}^2)}{\hat{\gamma} + \pi(\frac{x_n^2}{4} + \hat{\gamma}^2)}\,, \tag{111}$$

$$\pi\widehat{f}_-(0) = \sum_{|x_n|<\delta} \frac{\pi(\frac{x_n^2}{4} + \hat{\gamma}^2)}{\hat{\gamma} + \pi(\frac{x_n^2}{4} + \hat{\gamma}^2)}\,, \tag{112}$$

where $x_n$ are solutions of the equation

$$\frac{x}{2}\sin\pi\left(\frac{x}{2} + r\right) - \hat{\gamma}\cos\pi\left(\frac{x}{2} + r\right) = 0\,, \tag{113}$$

and $\hat{\gamma}, r$ are defined by

$$0 < \{\delta/2\} < \frac{1}{2} : \quad \hat{\gamma} = \frac{\delta}{2}\tan\left(\frac{\pi\delta}{2}\right) > 0 \, , \ r = 0 \, , \tag{114}$$

$$\frac{1}{2} < \{\delta/2\} < 1 : \quad \hat{\gamma} = -\frac{\delta}{2}\cot\left(\frac{\pi\delta}{2}\right) > 0 \, , \ r = \frac{1}{2} \, . \tag{115}$$

## 7 Virasoro primaries

Now we generalize to counting Virasoro primaries in $c > 1$ theories. The discussion here is essentially a refinement of the section 6 in [25] and similar to the section 2 of the present work, so we just highlight some key equations that have new features.

First, we consider bounds on the number of Virasoro primaries of all spins. The reduced partition function with zero angular potential ($\tau = i\frac{\beta}{2\pi}$)

$$z(\beta) = |\eta(\tau)|^2 Z(\beta) = e^{\beta\frac{c-1}{12}}\left[(1 - e^{-\beta})^2 + \sum_{\Delta > 0} e^{-\beta\Delta}\right] \tag{116}$$

is S-covariant

$$z(\beta) = \frac{2\pi}{\beta}z\left(\frac{4\pi^2}{\beta}\right) \, . \tag{117}$$

There are two new features here in comparison with section 2. First, there is a negative term in the RHS of (116) originating from the null states in the Virasoro vacuum module. Second, there is a power prefactor $\frac{2\pi}{\beta}$ in (117). We will show that these changes do not significantly alter the results. The negative term in (116) is not important because we are interested only in the tails of the sum. The role of the extra prefactor in (117) will be to change the definition of $\rho_0(\Delta)$.

It is convenient to make the following definitions

$$z(\beta) = z_{vac}(\beta) + z_{excited}(\beta) \, , \tag{118}$$

$$z_{vac}(\beta) = e^{\beta\frac{C}{12}}(1 - e^{-\beta})^2 \, , \tag{119}$$

$$z_{excited}(\beta) = \int_0^\infty d\Delta' \, \rho^{Vir}(\Delta')e^{-\beta(\Delta' - C/12)} \, , \tag{120}$$

where $C = c - 1$ and $\rho^{Vir}(\Delta')$ is the density of Virasoro primaries excluding the vacuum. Since $\rho^{Vir}$ is positive definite, we can use it in the arguments of the section 2 that led to bounds (13) and obtain

$$e^{\beta(\Delta - \delta - C/12)}\int_{-\Lambda}^{\Lambda} dt \, z_{excited}(\beta + it)\widehat{\phi}_-(t)e^{-itC/12} \leq \int_{\Delta - \delta}^{\Delta + \delta} d\Delta'\rho^{Vir}(\Delta') \tag{121}$$

$$\leq e^{\beta(\Delta + \delta - C/12)}\int_{-\Lambda}^{\Lambda} dt \, z_{excited}(\beta + it)\widehat{\phi}_+(t)e^{-itC/12} \, .$$

Now we take the limit $\Delta \to \infty, \beta \to 0$. The integrals in the LHS and RHS of (121) get large contributions of order $e^{\frac{4\pi}{\beta}\frac{C}{12}}$ from the region near $t = 0$. Therefore, we can add $z_{vac}(\beta + it)$, that never becomes exponentially large in the integration region $|t| < \Lambda$, under the $t$-integrals

without changing the asymptotic behavior of the bounds. The net effect is to substitute $z_{excited}(\beta + it)$ by $z(\beta + it)$ in (121)

$$e^{\beta(\Delta - \delta - C/12)} \int_{-\Lambda}^{\Lambda} dt \, z(\beta + it) \widehat{\phi}_-(t) e^{-itC/12} \leq \int_{\Delta - \delta}^{\Delta + \delta} d\Delta' \rho^{Vir}(\Delta') \qquad (122)$$

$$\leq e^{\beta(\Delta + \delta - C/12)} \int_{-\Lambda}^{\Lambda} dt \, z(\beta + it) \widehat{\phi}_+(t) e^{-itC/12} \, .$$

Then, following the argument in section 2, we S-dualize and split into light and heavy operators

$$z(\beta + it) = \frac{2\pi}{\beta + it} z_L \left( \frac{4\pi^2}{\beta + it} \right) + \frac{2\pi}{\beta + it} z_H \left( \frac{4\pi^2}{\beta + it} \right), \qquad (123)$$

where

$$z_L(\beta) = z_{vac}(\beta) + \sum_{0 < \Delta < C/12} e^{-\beta(\Delta - C/12)} \, , \qquad z_H(\beta) = \sum_{\Delta \geq C/12} e^{-\beta(\Delta - C/12)} \, . \qquad (124)$$

For light operators $z_L$ the $t$-integral is dominated by $t = 0$ and, in comparison with (19), the first term in the RHS of (123) contirubutes an extra factor $\frac{2\pi}{\beta + it} \to \frac{2\pi}{\beta}$. For heavy operators the $t$-integral is dominated by $t = \Lambda$ and we have an estimate

$$\left| z_H \left( \frac{4\pi^2}{\beta + it} \right) \right| \leq z_H \left( \frac{4\pi^2 \beta}{\beta^2 + t^2} \right) \sim \frac{t^2}{\beta} e^{\frac{t^2}{\beta} \frac{C}{12}} \, . \qquad (125)$$

Near $t \sim \Lambda$ this leads to an extra factor of $\frac{1}{\beta}$. Therefore, we have for the upper bound

$$\int_{\Delta - \delta}^{\Delta + \delta} d\Delta' \rho^{Vir}(\Delta') \leq \frac{2\pi}{\beta} \times \sqrt{\frac{3}{\pi c}} \beta^{3/2} e^{\beta\Delta + \frac{4\pi^2}{\beta} \frac{c}{12}} \widehat{\phi}_+(0) + O\left( \frac{1}{\beta} \times \beta^2 e^{\beta\Delta + \frac{\Lambda^2}{\beta} \frac{c}{12}} \right) . \qquad (126)$$

Both terms in the RHS get extra factors proportional to $\frac{1}{\beta}$ in comparison with (20). This shows that we can again choose $\Lambda = 2\pi$ and drop the second term in the RHS of (126) corresponding to heavy operators. Similar statements apply to the lower bound.

Finally, also optimizing over $\beta$, we have a generalization of (21) to Virasoro primaries at asymptotically large $\Delta$

$$2\pi \widehat{\phi}_-(0) \rho_0^{Vir}(\Delta) \leq \int_{\Delta - \delta}^{\Delta + \delta} d\Delta' \rho^{Vir}(\Delta') \leq 2\pi \widehat{\phi}_+(0) \rho_0^{Vir}(\Delta) \, , \qquad (127)$$

$$\rho_0(\Delta) = \left( \frac{3}{(c-1)\Delta} \right)^{1/4} \exp\left( 2\pi \sqrt{\frac{(c-1)\Delta}{3}} \right) . \qquad (128)$$

The functions $\widehat{\phi}_\pm(t)$ with support $|t| < \Lambda = 2\pi$ are chosen as described in sections 3 - 5 and summarized in the figure 3. The net change in comparison with (21) is to shift $c \to c - 1$ and multiply $\rho_0(\Delta)$ by $\frac{2\pi}{\beta}$ with $\beta = \pi\sqrt{\frac{c-1}{3\Delta}}$.

Similarly, one can repeat the arguments for fixed spin operators in section 6. The change is again to shift $c \to c - 1$ and multiply $\rho_J^0(\Delta)$ by $\frac{2\pi}{\beta}$ with $\beta = \pi\sqrt{\frac{c-1}{3\Delta}}$. Therefore, instead of (105), for Virasoro primaries of fixed spin $J$ we have

$$\pi \widehat{f}_-(0) \rho_J^{0,Vir}(\Delta) \leq \int_{\Delta - \delta}^{\Delta + \delta} d\Delta' \rho_J^{Vir}(\Delta') \leq \pi \widehat{f}_+(0) \rho_J^{0,Vir}(\Delta) \, , \qquad (129)$$

$$\rho_J^{0,Vir}(\Delta) = \left( \frac{2}{1 + \delta_{J,0}} \right) \frac{1}{\Delta} \exp\left[ 2\pi \sqrt{\frac{(c-1)\Delta}{3}} \right] . \qquad (130)$$

The functions $\widehat{f}_\pm(t)$ supported on $|t| < \pi$ can be chosen as described in section 6 with zero modes given either by (109) or (111), (112).

## Acknowledgements

We would like to thank D.Gorbachev and S.Tikhonov for pointing out the works of Selberg and Littmann. The work of BM is supported by NSF grant PHY-1911298. The work of SP is supported by Ambrose Monell Foundation and DOE grant DE-SC0009988.

## A    Majorization and Minorization by $\phi_{\pm}$

In this appendix, we show that $\phi_{\pm}(x)$ majorize and minorize the indicator function of the interval $[-\delta, \delta]$ for $2\delta \in \mathbb{Z}_+$, given the inequality (40) is satisfied.

In order to treat $\delta \in \mathbb{Z}$ and $\delta \in \mathbb{Z} + \frac{1}{2}$ cases simultaneously, we define functions $F_{\pm}(y) = \phi_{\pm}(y-\delta)$

$$
\begin{aligned}
F_+(y) &= \frac{\sin^2(\pi y)}{\pi^2}\left[\sum_{n=0}^{2\delta}\frac{1}{(y-n)^2} + \frac{\lambda_+}{y} + \frac{\lambda_+}{2\delta - y}\right], \\
F_-(y) &= \frac{\sin^2(\pi y)}{\pi^2}\left[\sum_{n=1}^{2\delta-1}\frac{1}{(y-n)^2} + \frac{\lambda_-}{y} + \frac{\lambda_-}{2\delta - y}\right].
\end{aligned}
\tag{131}
$$

We want to show that if the inequalities (40) are satisfied, we have

$$
F_-(y) \le \theta_{[0,2\delta]} \le F_+(y).
$$

We will be explicitly doing the analysis for $F_-(y)$ below. The analysis for $F_+(y)$ is similar. There are two regions of interest. For $y \in (0, 2\delta)$, we want $F_-(y) - 1 \le 0$ and for $y \notin (0, 2\delta)$, we want $F_-(y) \le 0$. We will see that requiring the former gives the upper bound on $\lambda_-$, while requiring the latter provides us with the lower bound on $\lambda_-$.

- $y \in (0, 2\delta)$: Using the identity

$$
1 = \frac{\sin^2(\pi y)}{\pi^2}\sum_{n=-\infty}^{\infty}\left(\frac{1}{y-n}\right)^2,
$$

we have

$$
F_-(y) - 1 = \frac{\sin^2(\pi y)}{\pi^2}\left[-\sum_{n=0}^{\infty}\left(\frac{1}{(y+n)^2} + \frac{1}{(2\delta+n-y)^2}\right) + \frac{\lambda_-}{y} + \frac{\lambda_-}{2\delta - y}\right].
\tag{132}
$$

Thus, we require

$$
\begin{aligned}
\lambda_- &\le \frac{y(2\delta - y)}{2\delta}\sum_{n=0}^{\infty}\left(\frac{1}{(y+n)^2} + \frac{1}{(2\delta+n-y)^2}\right) \text{ for } y \in (0, 2\delta), \\
\Leftrightarrow \lambda_- &\le \underset{y \in (0,2\delta)}{\text{Min}}\left(\frac{y(2\delta - y)}{2\delta}\sum_{n=0}^{\infty}\left(\frac{1}{(y+n)^2} + \frac{1}{(2\delta+n-y)^2}\right)\right).
\end{aligned}
\tag{133}
$$

The quantity in the brackets is minimized for $y = \delta$, as we will show below. Therefore

$$
\lambda_- \le \delta\sum_{n=0}^{\infty}\frac{1}{(\delta+n)^2} = \delta\psi_1(\delta).
\tag{134}
$$

Finally, to show that the RHS of (133) is indeed minimized for $y = \delta$, following [30], we argue as follows. We wish to show that for $y \in (0, 2\delta)$

$$
\left(\frac{y(2\delta - y)}{2\delta}\sum_{n=0}^{\infty}\left(\frac{1}{(y+n)^2} + \frac{1}{(2\delta+n-y)^2}\right)\right) - \delta\sum_{n=0}^{\infty}\frac{1}{(\delta+n)^2} \ge 0.
$$

We multiply the L.H.S by $2\delta$ and write it as

$$L \equiv \underbrace{y(2\delta - y) \sum_{n=0}^{\infty} g(n)}_{Term\ I} - \underbrace{2(y - \delta)^2 \sum_{n=0}^{\infty} \frac{1}{(\delta + n)^2}}_{Term\ II} ,$$

where

$$g(n) \equiv \left( \frac{1}{(y + n)^2} + \frac{1}{(2\delta + n - y)^2} - \frac{2}{(\delta + n)^2} \right).$$

Now the idea is to put a lower bound on term $I$ and an upper bound on term $II$, such that the lower bound on term $I$ is still bigger than the upper bound on term $II$, resulting in $L \geq 0$, which we want to prove.

⋆ Term $I$: Since the function $g$ has positive second derivative, the trapezoidal rule always overestimates the integral below and we have

$$\frac{g(n) + g(n + 1)}{2} \geq \int_n^{n+1} dx\ g(x), \tag{135}$$

which, upon summing over $n$, gives us

$$\sum_{n=0}^{\infty} g(n) \geq \frac{1}{2} g(0) + \int_0^{\infty} dx\ g(x). \tag{136}$$

⋆ Term $II$: We note that

$$\sum_{n=0}^{\infty} \frac{1}{(\delta + n)^2} = \frac{1}{\delta^2} + \sum_{n=1}^{\infty} \frac{1}{(\delta + n)^2} \leq \frac{1}{\delta^2} + \int_{1/2}^{\infty} dx\ g(x) = \frac{1}{\delta^2} + \frac{2}{2\delta + 1}, \tag{137}$$

where we have used the Jensen's inequality $g(k) \leq \int_{k-1/2}^{k+1/2} dn\ g(n)$ as $g$ is a positive, decreasing function of $n$ with positive second derivative.

⋆ Term $I$+ Term $II$: Combining eq. (136) and eq. (137), we have for $y \in (0, 2\delta)$

$$L \geq \frac{y(2\delta - y)}{2} g(0) - \frac{2(y - \delta)^2}{\delta^2} + y(2\delta - y) \int_0^{\infty} dx\ g(x) - \frac{4(y - \delta)^2}{2\delta + 1}$$
$$= \frac{(y - \delta)^2 \left( \delta^2 + (y - \delta)^2 \right)}{\delta^2 y(2\delta - y)} + \frac{2(y - \delta)^2}{\delta(2\delta + 1)} \geq 0. \tag{138}$$

• $y \notin (0, 2\delta)$: By construction we have $F_-(0) = F_-(2\delta) = 0$. Thus we need to consider $y \notin [0, 2\delta]$. Since $F_-(y)$ is symmetric around $\delta$, considering the function for $y < 0$ suffices. By symmetry, it is related to $y > 2\delta$. Let us focus on $y < 0$ and use the variable $w = -y$. We want to show that

$$\lambda_- \geq 1 - \frac{1}{2\delta} \Rightarrow \sum_{n=1}^{2\delta - 1} \frac{1}{(n + w)^2} - \frac{2\delta \lambda_-}{w(2\delta + w)} \leq 0 \text{ for } w > 0.$$

To prove this, we first note that

$$\frac{1}{(n + w)^2} \leq \frac{n}{w(n + 1 + w)} - \frac{n - 1}{w(n + w)} \text{ for } w > 0, n \geq 1$$
$$\Rightarrow \sum_{n=1}^{2\delta - 1} \frac{1}{(n + w)^2} \leq \frac{2\delta - 1}{w(2\delta + w)}. \tag{139}$$

Thus, we have

$$\sum_{n=1}^{2\delta-1} \frac{1}{(n+w)^2} - \frac{2\delta\lambda_-}{w(2\delta+w)} \leq \sum_{n=1}^{2\delta-1} \frac{1}{(n+w)^2} - \frac{2\delta-1}{w(2\delta+w)} \leq 0 \,. \tag{140}$$

We also remark that if $F_-(y) \leq 0$ for $y \notin [0, 2\delta]$, we can consider $y^2 F_-(y)$ and take $y \to \infty$ limit to deduce $\lambda_- \geq 1 - \frac{1}{2\delta}$. Thus the inequality implies and is implied by $F_-(y) \leq 0$ for $y \notin [0, 2\delta]$.

# B  Degeneracy of states for integer spaced spectra

In this section we use Ingham's theorem [3, 25, 41] to derive an asymptotic formula for the degeneracy of states $d_k$ in CFTs with integer spectra $\Delta_k = k \in \mathbb{Z}$. We assume that degeneracies of operators are non-decreasing $d_{k+1} \geq d_k$ and consider a modular invariant partition function

$$Z(\beta) = e^{\frac{\beta c}{12}} \sum_{k=0}^{\infty} d_k e^{-\beta k} \,. \tag{141}$$

Let us introduce an auxiliary function [41]

$$F(\beta) = (1 - e^{-\beta}) Z(\beta) = e^{\frac{\beta c}{12}} \sum_{k=0}^{\infty} (d_k - d_{k-1}) e^{-\beta k} \,,$$

where by definition $d_{-1} = 0$. In the $\beta \to 0$ limit we have

$$F(\beta) \underset{\beta \to 0}{\simeq} \beta e^{\frac{\pi^2 c}{3\beta}} \,.$$

The Ingham's theorem [3, 25, 41] implies that

$$d_N = \sum_{k=0}^{N} (d_k - d_{k-1}) \underset{N \to \infty}{\simeq} \rho_0(N) \,, \tag{142}$$

where $\rho_0$ is defined in eq. (22).

To apply (142) to partition functions considered in § 3.3 we need to check that $d_{k+1} \geq d_k$ is satisfied. First, it's easy to check the following statement. Consider two series expansions, one with non-decreasing and one with non-negative degeneracies

$$A(q) = \sum_{n=0}^{\infty} a_n q^n, \qquad a_{n+1} \geq a_n \geq 0 \,, \tag{143}$$

$$B(q) = \sum_{n=0}^{\infty} b_n q^n, \qquad b_n \geq 0 \,. \tag{144}$$

Then the product $AB$ gives rise to non-decreasing degeneracies $c_N$

$$A(q)B(q) = \sum_{N=0}^{\infty} c_N q^N \,, \tag{145}$$

$$c_N = \sum_{n=0}^{N} a_{N-n} b_n \,. \tag{146}$$

Indeed, one simply compares term by term

$$c_{N+1} = \sum_{n=0}^{N+1} a_{N+1-n} b_n \geq \sum_{n=0}^{N} a_{N-n} b_n = c_N \ . \tag{147}$$

Now let's show that $j(\tau)^{1/3}$ has non-decreasing expansion coefficients. We recall that

$$j(\tau)^{1/3} = \frac{1 + 240 \sum_n \sigma_3(n) q^n}{\eta(\tau)^8} \ , \quad q = e^{2\pi i \tau} \ .$$

Since $\frac{1}{\eta(\tau)}$ has non-decreasing expansion coefficients, by the general statement above $j(\tau)^{1/3}$ gives rise to non-decreasing degeneracies $d_{k+1} \geq d_k$. Similarly, any integer power of $j(\tau)^{1/3}$ satisfies $d_{k+1} \geq d_k$.

## C  Zero mode of Beurling-Selberg function

The computation in this section is from [30]. We recall

$$B_{\pm}(x) = \frac{\sin^2(\pi x)}{\pi^2} \left[ \frac{2}{x} \pm \frac{1}{x^2} + \sum_{k=1}^{\infty} \left( \frac{1}{(x-k)^2} - \frac{1}{(x+k)^2} \right) \right] \ . \tag{148}$$

We wish to show that

$$\int_{-\infty}^{\infty} dx \left( \frac{1}{2} B_{\pm}(\delta+x) + \frac{1}{2} B_{\pm}(\delta-x) \right) = (2\delta \pm 1) \ . \tag{149}$$

Equivalently

$$\frac{1}{2} \int_{-\infty}^{\infty} dx \left( B_{\pm}(\delta+x) - \text{sign}(\delta+x) + B_{\pm}(\delta-x) - \text{sign}(\delta-x) \right) = \pm 1 \ . \tag{150}$$

First note that

$$\int_{-\infty}^{\infty} dx \left( B_{\pm}(x) - \text{sign}(x) \right) \quad \text{is finite} \ .$$

This follows from

$$1 = \frac{\sin^2(\pi x)}{\pi^2} \sum_{k \in \mathbb{Z}} \frac{1}{(x-k)^2} \tag{151}$$

and

$$\begin{aligned} B_+(x) - \text{sgn}(x) &= \frac{\sin^2(\pi x)}{\pi^2} \left[ \frac{2}{x} - 2\psi_1(1+x) \right] \underset{|x| \to \infty}{\sim} \frac{\sin^2(\pi x)}{\pi^2 x^2} \ , \\ B_-(x) - \text{sgn}(x) &= \frac{\sin^2(\pi x)}{\pi^2} \left[ \frac{2}{x} - 2\psi_1(x) \right] \underset{|x| \to \infty}{\sim} -\frac{\sin^2(\pi x)}{\pi^2 x^2} \ . \end{aligned} \tag{152}$$

Now, by shifting $x$, we have

$$\frac{1}{2} \int_{-\infty}^{\infty} dx \left( B_{\pm}(\delta+x) - \text{sign}(\delta+x) + B_{\pm}(\delta-x) - \text{sign}(\delta-x) \right) \tag{153}$$

$$= \frac{1}{2} \int_{-\infty}^{\infty} dx \left( B_{\pm}(x) - \text{sign}(x) + B_{\pm}(-x) - \text{sign}(-x) \right) \tag{154}$$

$$= \frac{1}{2} \int_{-\infty}^{\infty} dx \left( B_{\pm}(x) + B_{\pm}(-x) \right) = \pm \int_{-\infty}^{\infty} dx \frac{\sin^2(\pi x)}{(\pi x)^2} = \pm 1 \ . \tag{155}$$

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
