# Peer review of "Beurling-Selberg Extremization and Modular Bootstrap at High Energies"

_SciPost Physics, doi:SciPost Phys. 8, 088 (2020)_

## Round 1 · Referee Report · Anonymous (Referee 1) · 2020-5-18

Strengths
- Places rigorous bounds on interesting quantities in CFT
- Written clearly and concisely, and defines the problem carefully
- Formalizes various intuitive expectations people had about CFT
- Makes interesting connections to previous results in the math literature
Weaknesses
- Mostly is a result on rational CFT. It is expected (but not proved) that for irrational CFTs the bounds can be vastly improved
- Mostly a followup of previous work in the literature (including previous work by both authors)
Report
The main technique this paper uses is by bounding a Heaviside theta function of width $2\delta$ centered at $\Delta$ by two smooth functions with Fourier transforms of finite support. By bounding the expressions for the density of states integrated against the lower and upper smooth functions, the authors are able to place rigorous upper and lower bounds on the number of operators in a certain window of size $2\delta$ centered on $\Delta$. It turns out that this problem of bounding such a theta function was previously explored by mathematicians and is known as the Beurling-Selberg extremization problem. By applying this technique to the CFT problem, the authors are able to provide stronger bounds on the number of operators at asymptotically large energy for any fixed width $\delta$. For example if the width $2\delta$ is less than $1$, then there is no lower bound of operators, which is reflected in the fact that there exist modular-invariant functions with support only at integer values of energy. In the limit as $\delta\rightarrow 0$, the authors find a nontrivial upper bound, which provides an interesting universal upper bound on the density of states at a particular energy $\Delta$.
This paper is clearly written and provides interesting results in the study of rational conformal field theory, and interesting connections to optimization problems in analytic number theory. I recommend that it should be published, with very minor modifications below.
Requested changes
- Very minor typos. On p. 3, "loose" $\rightarrow$ "lose" (twice) On p. 10, "consdier" $\rightarrow$ "consider" On p. 24, "degeneraices" $\rightarrow$ "degeneracies"
- When discussing the saturation by partition functions of $c=4k$ (e.g. in the Introduction and in Sec $3.3$), it seems to me that there are not necessarily $SL(2, \mathbb Z)$ invariant partition functions that saturate these bounds if $c = 4 ~\text{mod}~ 8$. Rather it seems that there are $S$-invariant functions that saturate these bounds (that do not necessarily consistently "split" amongst spins into an $SL(2, \mathbb Z)$ invariant function). Is this correct? If so, then since for the majority of the paper the authors only use $S$-invariance, it would be good to clarify this point.
- In (7) should we expect $\delta$ dependence on the RHS, i.e. should the RHS be something like $2\delta e^{2\pi \sqrt{\frac{c\Delta}3}}$?
- Most of the paper is about the case of states rather than Virasoro primaries, where the result is technically trivial. The authors write that a minor modification of the argument will go through for the case of Virasoro primaries. If it is not too difficult it may be useful to expand on this point in more detail. It seems to me that two qualitative features may change when one considers Virasoro primaries if $c>1$: (1) The modular crossing equation will pick up a "weight" coming from the Virasoro character transforming as a modular form of weight $-\frac12$. (2) There are null states in representations of the Virasoro vacuum character which will effect vacuum states and chiral operators. Although intuitively one might expect that these two changes will not affect the main result, it would be good to explain these more rigorously if possible, since the main result of this paper is a careful rigorous derivation of expected-results in CFT.
Author: Baurzhan Mukhametzhanov on 2020-05-22 [id 836]
(in reply to Report 1 on 2020-05-18)
Dear Referee,
thank you for your report. We've tried to address all requested changes that you made in the new version (to be submitted shortly). Here we would like to make a few comments about those changes and the report itself.
2) You are right, in section 3.3 we consider partition functions that are S-invariant, but not necessarily SL(2,Z) invariant. This is because we consider $j(\tau)$ as a non-holomorphic partition function with $\tau = -\bar \tau = i\beta/2\pi$. We clarified it in the new version.
3) Yes, it is natural to expect that the number of states in an interval is proportional to its size $2\delta$. In (7) we just wanted to emphasize the expectation about the exponent. At this point we can’t be precise about the prefactor, as it contains other factors besides $2\delta$. We’ve added “#” in (7) to indicate that this part is to be made more precise later in the paper.
4) We’ve added section 7 on Virasoro primaries. We explicitly discuss there why the two changes that you mentioned do not significantly alter the results. Essentially, the point is that null states of the vacuum character are not important because we are interested only in the tails of the partition function. And the effect of extra factors of $\beta$ in modular crossing is to change the definition of $\rho_0$.
Comments on the Report:
We agree with your description “Mostly is a result on rational CFT” in the sense that only RCFTs are close to saturating (and sometimes do saturate) our bounds. We don’t see however why is this counted as a weakness. Our results are about what is universal in all 2d CFTs, even though they are expected to be significantly improved in irrational CFTs. It is a separate very interesting question how by making extra assumptions one can improve the bounds and see the expected features of irrational CFTs.
Similarly, we don’t quite agree with the adjective “rational” in “This paper…provides interesting results in the study of rational conformal field theory” since our results are universal for all 2d unitary CFTs, both rational and irrational.
Thank you, Baur and Sridip.
Author: Baurzhan Mukhametzhanov on 2020-05-22 [id 837]
(in reply to Report 2 by Slava Rychkov on 2020-05-19)Dear Slava,
thank you for the report. We've tried to address the requested changes you made in the new version (to be submitted shortly). A few remarks about your questions (the rest is addressed in the list of changes of the new version):
We’ve added the formula (9) and discussion around it to clarify the meaning of the limit $\Delta \to \infty$. The main point is that the quantity $N_\delta(\Delta) = \int_{\Delta - \delta}^{\Delta + \delta} d\Delta' \rho(\Delta')$ is a “staircase”-like function that might be oscillating around its average smooth function $\rho_0(\Delta)$ all the way to arbitrarily large $\Delta$. Therefore, $lim_{\Delta \to \infty} N_\delta(\Delta)/\rho_0(\Delta)$ might not exist and instead it is natural to consider $\limsup$ and $\liminf$, as we discuss in the new version. Hopefully this makes the meaning of (8) clearer.
Our bounds are only about the asymptotic values. Relaxing the constants by a finite $\epsilon$ and deriving the bounds for $\Delta \geq \Delta_0(\epsilon)$, as you suggested, would be very interesting. We suspect the exact value of $\Delta_0(\epsilon)$ to be theory dependent. But perhaps the growth of $\Delta_0(\epsilon)$ as $\epsilon \to 0$ might be universal. In any case this would require a more refined analysis of error terms in our arguments. We have not attempted that.
We've also tried to reduce the dependence on [25] as much as possible. We think now it is a self-contained discussion. We've also added a section about Virasoro.
Thank you,
Baur and Sridip.

---

## Round 1 · Referee Report · Slava Rychkov (Referee 2) · 2020-5-19

Strengths
1- establishes a useful connection between contemporary physics and classic mathematics literatures
Report
Requested changes
1- I found formulation of the main results of the paper somewhat vague. What does it mean that (8) is true for $\Delta\to\infty$? Does it mean that if I relax the constants in the r.h.s. and l.h.s. by an $\epsilon>0$ the bounds will be true for all $\Delta≥ \Delta_0(\epsilon)$? Do the authors's methods allow to determine how $\Delta_0$ depend on $\epsilon$, and if not why not?
2- It would be nice if the authors reduced the dependence of this work on [25]. As it stands, the reader not only should be familiar with the reasoning of [25], but also with the notation (e.g. $Z_H$ is not explicitly defined here). Furthermore, the reasoning seems somewhat different, e.g. the HKS bound does not seem to be mentioned in the paper under review; is it needed or used here, perhaps implicitly? If not why not? Several times the authors say: "See [25]", and a more pointed reference would be welcome.
3- At several places (p.4, p.20), the authors state that their arguments apply also to Virasoro primaries, with the same results. I think it would be nice to at least state those results (even if they have a very similar formulation, as this will facilitate future referencing), and provide a short explanation (at least in words if not in equations) what are the changes needed and why they don't invalidate the arguments. As it stands, the readers not perfectly familiar with [25] may be left incredulous.
More minor remarks
4- p. 3 looses -> loses
5 - p.17, Eq. (89), second equation, should $\tau$ be $\bar\tau$?

---

## Round 3 · List of Changes

1) Formulas (9), (10) and discussion around them is added to clarify the precise meaning of the formula (8) and its analogs. 2)"#" is added in the RHS of (7) to emphasize that the prefactor is to be made precise later in the paper. 3) Formula (12) is added and footnote 6 is modified to reduce the dependence of this work on [25]. 4) Paragraph after (17) is slightly modified to make the discussion more explicit and less dependent on [25]. 5) Formula (16) is added to give explicit definition of Z_H and Z_L and reduce dependence on [25]. 6) Paragraph before (18) is added about HKS bound. It was said in [25] that one can use HKS bound. Here we emphasize that it is not necessary and only high-temperature asymptotic of the partition function is needed. 7) Paragraph after (47) is modified to emphasize that we consider Klein's j-function as a non-holomorphic partition function that is S-invariant, but not necessarily $SL(2,Z)$ invariant. 8) Reference [42] is added after (50). 9) Section 7 about Virasoro primaries is added to make claims about this generalization more explicit and reduce dependence on [25]. 10) Typos pointed out in both reports are fixed.

---

## Editorial Decision

published